# Combinatorial Energy Learning for Image Segmentation

**Jeremy Maitin-Shepard**
UC Berkeley    Google
jbms@google.com

**Viren Jain**
Google
viren@google.com

**Michal Januszewski**
Google
mjanusz@google.com

**Peter Li**
Google
phli@google.com

**Pieter Abbeel**
UC Berkeley
pabbeel@cs.berkeley.edu

## Abstract

We introduce a new machine learning approach for image segmentation that uses a neural network to model the conditional energy of a segmentation given an image. Our approach, *combinatorial energy learning for image segmentation (CELIS)* places a particular emphasis on modeling the inherent combinatorial nature of dense image segmentation problems. We propose efficient algorithms for learning deep neural networks to model the energy function, and for local optimization of this energy in the space of supervoxel agglomerations. We extensively evaluate our method on a publicly available 3-D microscopy dataset with 25 billion voxels of ground truth data. On an 11 billion voxel test set, we find that our method improves volumetric reconstruction accuracy by more than 20% as compared to two state-of-the-art baseline methods: graph-based segmentation of the output of a 3-D convolutional neural network trained to predict boundaries, as well as a random forest classifier trained to agglomerate supervoxels that were generated by a 3-D convolutional neural network.

## 1   Introduction

Mapping neuroanatomy, in the pursuit of linking hypothesized computational models consistent with observed functions to the actual physical structures, is a long-standing fundamental problem in neuroscience. One primary interest is in mapping the network structure of neural circuits by identifying the morphology of each neuron and the locations of synaptic connections between neurons, a field called connectomics. Currently, the most promising approach for obtaining such maps of neural circuit structure is volume electron microscopy of a stained and fixed block of tissue. [4, 16, 17, 10] This technique was first used successfully decades ago in mapping the structure of the complete nervous system of the 302-neuron *Caenorhabditis elegans*; due to the need to manually cut, image, align, and trace all neuronal processes in about 8000 50 nm serial sections, even this small circuit required over 10 years of labor, much of it spent on image analysis. [31] At the time, scaling this approach to larger circuits was not practical.

Recent advances in volume electron microscopy [11, 20, 15] make feasible the imaging of large circuits, potentially containing hundreds of thousands of neurons, at sufficient resolution to discern even the smallest neuronal processes. [4, 16, 17, 10] The high image quality and near-isotropic resolution achievable with these methods enables the resultant data to be treated as a true 3-D volume, which significantly aids reconstruction of processes that do not run parallel to the sectioning axis, and is potentially more amenable to automated image processing.

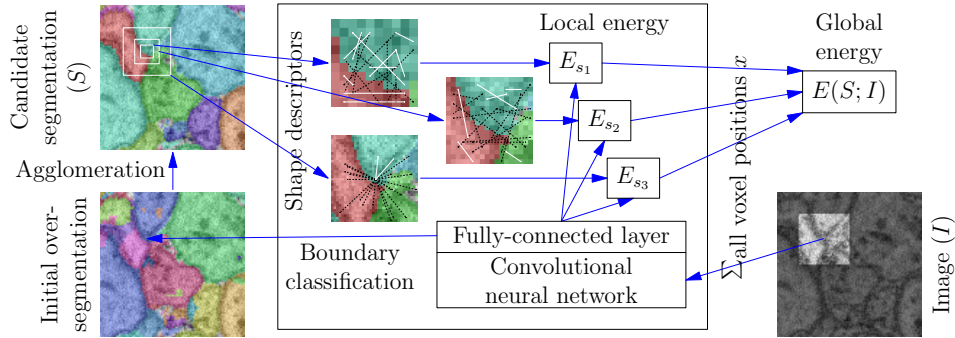

Figure 1: Illustration of computation of global energy for a *single* candidate segmentation $S$. The local energy $E_s(x; S; I) \in [0, 1]$, computed by a deep neural network, is summed over all shape descriptor types $s$ and voxel positions $x$.

Image analysis remains a key challenge, however. The primary bottleneck is in segmenting the full volume, which is filled almost entirely by heavily intertwined neuronal processes, into the volumes occupied by each individual neuron. While the cell boundaries shown by the stain provide a strong visual cue in most cases, neurons can extend for tens of centimeters in path length while in some places becoming as narrow as $40\,\mathrm{nm}$; a single mistake anywhere along the path can render connectivity information for the neuron largely inaccurate. Existing automated and semi-automated segmentation methods do not sufficiently reduce the amount of human labor required: a recent reconstruction of 950 neurons in the mouse retina required over 20000 hours of human labor, even with an efficient method of tracing just a skeleton of each neuron [18]; a recent reconstruction of 379 neurons in the *Drosophila* medulla column (part of the visual pathway) required 12940 hours of manual proof-reading/correction of an automated segmentation [26].

**Related work:** Algorithmic approaches to image segmentation are often formulated as variations on the following pipeline: a boundary detection step establishes local hypotheses of object boundaries, a region formation step integrates boundary evidence into local regions (i.e. superpixels or supervoxels), and a region agglomeration step merges adjacent regions based on image and object features. [1, 19, 30, 2] Although extensive integration of machine learning into such pipelines has begun to yield promising segmentation results [3, 14, 22], we argue that such pipelines, as previously formulated, fundamentally neglect two potentially important aspects of achieving accurate segmentation: (i) the combinatorial nature of reasoning about dense image segmentation structure,[1] and (ii) the fundamental importance of shape as a criterion for segmentation quality.

**Contributions:** We propose a method that attempts to overcome these deficiencies. In particular, we propose an energy-based model that scores segmentation quality using a deep neural network that flexibly integrates shape and image information: Combinatorial Energy Learning for Image Segmentation (CELIS). In pursuit of such a model this paper makes several specific contributions: a novel connectivity region data structure for efficiently computing the energy of configurations of 3-D objects; a binary shape descriptor for efficient representation of 3-D shape configurations; a neural network architecture that splices the intermediate unit output from a trained convolutional network as input to a deep fully-connected neural network architecture that scores a segmentation and 3-D image; a training procedure that uses pairwise object relations within a segmentation to learn the energy-based model. an experimental evaluation of the proposed and baseline automated reconstruction methods on a massive and (to our knowledge) unprecedented scale that reflects the true size of connectomic datasets required for biological analysis (many billions of voxels).

## 2   Conditional energy modeling of segmentations given images

We define a global, translation-invariant energy model for predicting the cost of a complete segmentation $S$ given a corresponding image $I$. This cost can be seen as analogous to the negative

log-likelihood of the segmentation given the image, but we do not actually treat it probabilistically. Our goal is to define a model such that the true segmentation corresponding to a given image can be found by minimizing the cost; the energy can reflect both a prior over object configurations alone, as well as compatibility between object configurations and the image.

As shown in Fig. 1, we define the global energy $E(S; I)$ as the sum over local energy models (defined by a deep neural network) $E_s(x; S; I)$ at several different scales $s$ computed in sliding-window fashion centered at every position $x$ within the volume:

$$E(S; I) \coloneqq \sum_s \sum_x E_s(x; S; I),$$

$$E_s(x; S; I) \coloneqq \hat{E}_s\left(r_s(x; S); \phi(x; I)\right).$$

The local energy $E_s(x; S; I)$ depends on the local image context centered at position $x$ by way of a vector representation $\phi(x; I)$ computed by a deep convolutional neural network, and on the local shape/object configuration at scale $s$ by way of a novel *local binary shape descriptor* $r_s(x; S)$, defined in Section 3.

To find (locally) minimal-cost segmentations under this model, we use local search over the space of agglomerations starting from some initial supervoxel segmentation. Using a simple greedy policy, at each step we consider all possible agglomeration actions, i.e. merges between any two adjacent segments, and pick the action that results in the lowest energy.

Naïvely, computing the energy for just a single segmentation requires computing shape descriptors and then evaluating the energy model at every voxel position with the volume; a small volume may have tens or hundreds of millions of voxels. At each stage of the agglomeration, there may be thousands, or tens of thousands, of potential next agglomeration steps, each of which results in a unique segmentation. In order to choose the best next step, we must know the energy of all of these potential next segmentations. The computational cost to perform these computations *directly* would be tremendous, but in the supplement, we prove a collection of theorems that allow for an efficient implementation that computes these energy terms *incrementally*.

## 3 Representing 3-D Shape Configurations with Local Binary Descriptors

We propose a *binary shape descriptor* based on subsampled pairwise connectivity information: given a specification $s$ of $k$ pairs of position offsets $\{a_1, b_1\}, \ldots, \{a_k, b_k\}$ relative to the center of some fixed-size bounding box of size $B_s$, the corresponding $k$-bit binary shape descriptor $r(U)$ for a particular segmentation $U$ of that bounding box is defined by

$$r^i(U) \coloneqq \begin{cases} 1 & \text{if } a_i \text{ is connected to } b_i \text{ in } U; \\ 0 & \text{otherwise.} \end{cases} \qquad \text{for } i \in [1, k].$$

As shown in Fig. 2a, each bit of the descriptor specifies whether a particular pair of positions are part of the same segment, which can be determined in constant time by the use of a suitable data structure. In the limit case, if we use the list of all $\binom{n}{2}$ pairs of positions within an $n$-voxel bounding box, no information is lost and the Hamming distance between two descriptors is precisely equal to the Rand index. [23] In general we can sample a subset of only $k$ pairs out of the $\binom{n}{2}$ possible; if we sample uniformly at random, we retain the property that the *expected* Hamming distance between two descriptors is equal to the Rand index. We found that picking $k = 512$ bits provides a reasonable trade-off between fidelity and representation size. While the pairs may be randomly sampled initially, naturally to obtain consistent results when learning models based on these descriptors we must use the same fixed list of positions for defining the descriptor at both training and test time. [2]

Note that this descriptor serves in general as a type of sketch of a full segmentation of a given bounding box. By restricting one of the two positions of each pair to be the center position of the bounding box, we instead obtain a sketch of just the single segment containing the center position. We refer to the descriptor in this case as *center-based*, and to the general case as *pairwise*, as shown in Fig. 2b. We will use these shape descriptors to represent only *local sub-regions* of a segmentation. To represent shape information throughout a large volume, we compute shape descriptors densely at all positions in a sliding window fashion, as shown in Fig. 2c.



$r = 1\ldots$  $\qquad$  $r = 100000000110\ldots$  $\qquad$  $r = 10000000011000000110100000101001$

(a) Sequence showing computation of a shape descriptor.

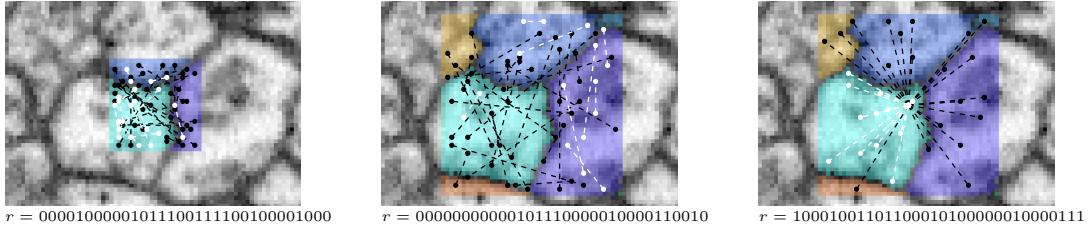

$r = 00001000001011100111100100001000$  $\quad$  $r = 00000000000101110000010000110010$  $\quad$  $r = 1000100110110001010000001000111$

(b) Shape descriptors are computed at multiple scales. Pairwise descriptors (shown left and center) consider arbitrary pairwise connectivity, while center-based shape descriptors (shown right) restrict one position of each pair to be the center point.

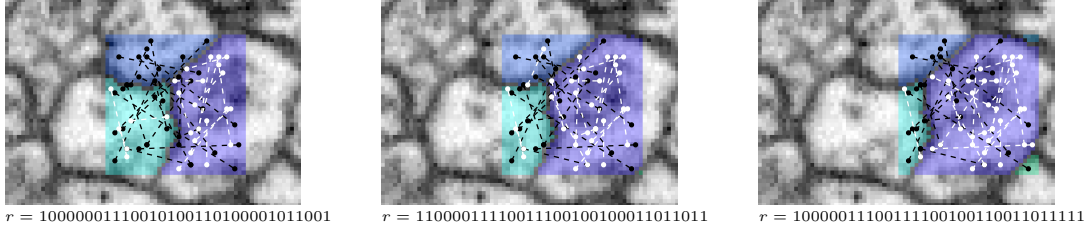

$r = 100000011100101001101000001011001$  $\quad$  $r = 110000111100111001001000110111011$  $\quad$  $r = 100000111001111001001100110111111$

(c) Shape descriptors are computed densely at every position within the volume.

Figure 2: Illustration of shape descriptors. The connected components of the bounding box $U$ for which the descriptor is computed are shown in distinct colors. The pairwise connectivity relationships that define the descriptor are indicated by dashed lines; connected pairs are shown in white, while disconnected pairs are shown in black. Connectivity is determined based on the connected components of the underlying segmentation, not the geometry of the line itself. While this illustration is 2-D, in our experiments shape descriptors are computed fully in 3-D.

**Connectivity Regions**

As defined, a single shape descriptor represents the segmentation within its fixed-size bounding box; by shifting the position of the bounding box we can obtain descriptors corresponding to different local regions of some larger segmentation. The size of the bounding box determines the *scale* of the local representation. This raises the question of how connectivity should be defined within these local regions. Two voxels may be connected only by a long path well outside the descriptor bounding box. As we would like the shape descriptors to be consistent with the local topology, such pairs should be considered disconnected. Shape descriptors are, therefore, defined with respect to connectivity within some larger *connectivity region*, which necessarily contains one or more descriptor bounding boxes but may in general be significantly smaller than the full segmentation; conceptually, the shape descriptor bounding box slides around to all possible positions contained within the connectivity region. (This sliding necessarily results in some minor inconsistency in context between different positions, but reduces computational and memory costs.) To obtain shape descriptors at all positions, we simply tile the space with overlapping rectangular connectivity regions of appropriate uniform size and stride, as shown in the supplement. The connectivity region size determines the degree of locality of the connectivity information captured by the shape descriptor (independent of the descriptor bounding box size). It also affects computational costs, as described in the supplement.

# 4 Energy model learning

We define the local energy model $\hat{E}_s(r; v)$ for each shape descriptor type/scale $s$ by a learned neural network model that computes a real-valued score in $[0, 1]$ from a shape descriptor $r$ and image feature vector $v$.

To simplify the presentation, we define the following notation for the forward discrete derivative of $f$ with respect to $S$: $\Delta_S^e f(S) \coloneqq f(S + e) - f(S)$.

Based on this notation, we have the discrete derivative of the energy function $\Delta_S^e E(S; I) = E(S + e; I) - E(S; I)$, where $S + e$ denotes the result of merging the two supervoxels corresponding to $e$ in the existing segmentation $S$. To agglomerate, our greedy policy simply chooses at step $t$ the action $e$ that minimizes $\Delta_{S^t}^e E(S^t; I)$, where $S^t$ denotes the current segmentation at step $t$.

As in prior work [22], we treat this as a classification problem, with the goal of matching the sign of $\Delta_{S^t}^e E(S^t; I)$ to $\Delta_{S^t}^e \mathrm{error}(S^t, S^*)$, the corresponding change in segmentation error with respect to a ground truth segmentation $S^*$, measured using Variation of Information [21].

## 4.1 Local training procedure

Because the $\Delta_{S^t}^e E(S^t; I)$ term is simply the sum of the change in energies from each position and descriptor type $s$, as a heuristic we optimize the parameters of the energy model $\hat{E}_s(r; v)$ independently for each shape descriptor type/scale $s$. We seek to minimize the expectation

$$\mathbb{E}_i \bigg[ \ell(\Delta_{S_i}^{e_i} \mathrm{error}(S_i, S^*), \hat{E}_s(r_s(x_i; S_i + e); \phi(x_i; I))) +$$

$$\ell(-\Delta_{S_i}^{e_i} \mathrm{error}(S_i, S^*), \hat{E}_s(r_s(x; S_i); \phi(x_i; I))) \bigg],$$

where $i$ indexes over training examples that correspond to a particular sampled position $x_i$ and a merge action $e_i$ applied to a segmentation $S_i$. $\ell(y, a)$ denotes a binary classification loss function, where $a \in [0, 1]$ is the predicted probability that the true label $y$ is positive, weighted by $|y|$. Note that if $\Delta_{S_i}^{e_i} \mathrm{error}(S_i, S^*) < 0$, then action $e$ improved the score and therefore we want a low predicted score for the post-merge descriptor $r_s(x_i; S_i + e)$ and a high predicted score for the pre-merge descriptor $r_s(x_i; S_i)$; if $\Delta_{S_i}^{e_i} \mathrm{error}(S_i, S^*) > 0$ the opposite applies. We tested the standard log loss $\ell(y, a) \coloneqq |y| \cdot [\mathbb{1}_{y>0} \log(a) + \mathbb{1}_{y<0} \log(1 - a)]$, as well as the signed linear loss $\ell(y, a) \coloneqq y \cdot a$, which more closely matches how the $E_s(x; S_i; I)$ terms contribute to the overall $\Delta_S^e E(S; I)$ scores. Stochastic gradient descent (SGD) is used to perform the optimization.

We obtain training examples by agglomerating using the *expert* policy that greedily optimizes $\mathrm{error}(S^t, S^*)$. At each segmentation state $S^t$ during an agglomeration step (including the initial state), for each possible agglomeration action $e$, and each position $x$ within the volume, we compute the shape descriptor pair $r_s(x; S^t)$ and $r_s(x; S^t + e)$ reflecting the pre-merge and post-merge states, respectively. If $r_s(x; S^t) \neq r_s(x; S^t + e)$, we emit a training example corresponding to this descriptor pair. We thereby obtain a conceptual stream of examples $\langle e, \Delta_{S^t}^e \mathrm{error}(S^t, S^*), \phi(x; I), r_s(x; S^t), r_s(x; S^t + e) \rangle$.

This stream of examples may contain billions of examples (and many highly correlated), far more than required to learn the parameters of $E_s$. To reduce resource requirements, we use priority sampling [12], based on $|\Delta_S^e \mathrm{error}(S, S^*)|$, to obtain a fixed number of weighted samples without replacement for each descriptor type $s$. We equalize the total weight of true merge examples ($\Delta_S^e \mathrm{error}(S, S^*) < 0$) and false merge examples ($\Delta_S^e \mathrm{error}(S, S^*) > 0$) in order to avoid learning degenerate models.[3]

# 5 Experiments

We tested our approach on a large, publicly available electron microscopy dataset, called *Janelia FIB-25*, of a portion of the *Drosophila melangaster* optic lobe. The dataset was collected at $8 \times 8 \times 8 \, \mathrm{nm}$

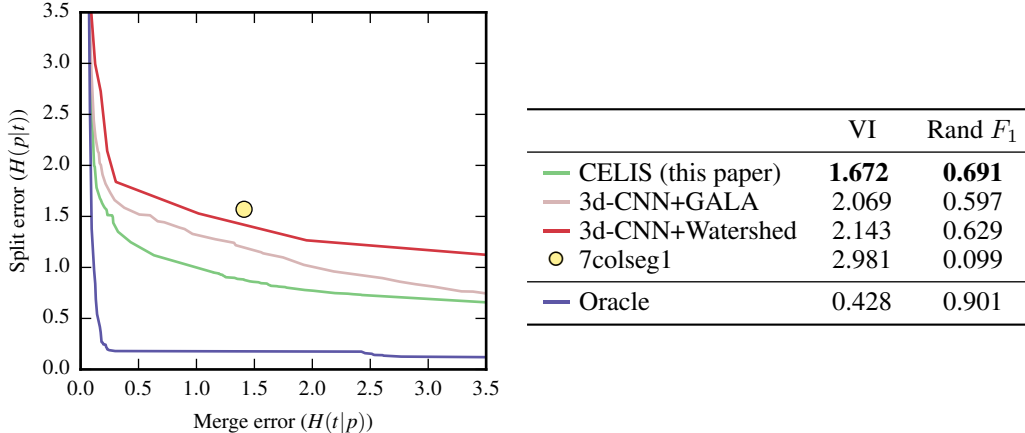

| | VI | Rand $F_1$ |
|---|---|---|
| CELIS (this paper) | **1.672** | **0.691** |
| 3d-CNN+GALA | 2.069 | 0.597 |
| 3d-CNN+Watershed | 2.143 | 0.629 |
| 7colseg1 | 2.981 | 0.099 |
| Oracle | 0.428 | 0.901 |

Figure 3: Segmentation accuracy on 11-gigavoxel FIB-25 test set. Left: Pareto frontiers of information-theoretic split/merge error, as used previously to evaluate segmentation accuracy. [22] Right: Comparison of Variation of Information (lower is better) and Rand $F_1$ score (higher is better). For CELIS, 3d-CNN+GALA, and 3d-CNN+watershed, the hyperparameters were optimized for each metric on the *training* set.

resolution using Focused Ion Beam Scanning Electron Microscopy (FIB-SEM); a labor-intensive semi-automated approach was used to segment all of the larger neuronal processes within a $\approx$ 20,000 cubic micron volume (comprising about 25 billion voxels). [27] To our knowledge, this challenging dataset is the largest publicly available electron microscopy dataset of neuropil with a corresponding "ground truth" segmentation.

For our experiments, we split the dataset into separate training and testing portions along the z axis: the training portion comprises z-sections 2005–5005, and the testing portion comprises z-sections 5005–8000 (about 11 billion voxels).

## 5.1 Boundary classification and oversegmentation

To obtain image features and an oversegmentation to use as input for agglomeration, we trained convolutional neural networks to predict, based on a $35 \times 35 \times 9$ voxel image context region, whether the center voxel is part of the same neurite as the adjacent voxel in each of the x, y, and z directions, as in prior work. [29] We optimized the parameters of the network using stochastic gradient descent with log loss. We trained several different networks, varying as hyperparameters the amount of dilation of boundaries in the training data (in order to increase extracellular space) from 0 to 8 voxels and whether components smaller than 10000 voxels were excluded. See the supplementary information for a description of the network architecture. Using these connection affinities, we applied a watershed algorithm [33, 34] to obtain an (approximate) oversegmentation. We used parameters $T_l = 0.95$, $T_h = 0.95$, $T_e = 0.5$, and $T_s = 1000$ voxels.

## 5.2 Energy model architecture

We used five types of 512-dimensional shape descriptors: three pairwise descriptor types with $9^3$, $17^3$, and $33^3$ bounding boxes, and two center-based descriptor types with $17^3$ and $33^3$ bounding boxes, respectively. The connectivity positions within the bounding boxes for each descriptor type were sampled uniformly at random.

We used the 512-dimensional fully-connected penultimate layer output of the low-level classification convolutional neural network as the image feature vector $\phi(x; I)$. For each shape descriptor type $s$, we used the following architecture for the local energy model $\hat{E}_s(r; v)$: we concatenated the shape descriptor vector and the image feature vector to obtain a 1024-dimensional input vector. We used two 2048-dimensional fully-connected rectified linear hidden layers, followed by a logistic output unit, and applied dropout (with $p = 0.5$) after the last hidden layer. While this effectively computes a

score from a raw image patch and a shape descriptor, by segregating expensive convolutional image processing that does not depend on the shape descriptor, this architecture allows us to benefit from pre-training and precomputation of the intermediate image feature vector $\phi(x; I)$ for each position $x$. Training for both the energy models and the boundary classifier was performed using asynchronous SGD using a distributed architecture. [9]

### 5.3 Evaluation

We compared our method to the state-of-the-art agglomeration method GALA [22], which trains a random forest classifier to predict merge decisions using image features derived from boundary probabilities. [4] To obtain such probabilities from our low-level convolutional neural network classifier, which predicts edge affinities *between* adjacent voxels rather than per-voxel predictions, we compute for each voxel the minimum connection probability to any voxel in its 6-connectivity neighborhood, and treat this as the probability/score of it being cell interior.

For comparison, we also evaluated a watershed procedure applied to the CNN affinity graph output, under varying parameter choices, to measure the accuracy of the deep CNN boundary classification without the use of an agglomeration procedure. Finally, we evaluated the accuracy of the publicly released automated segmentation of FIB-25 (referred to as *7colseg1*) [13] that was the basis of the proofreading process used to obtain the ground truth; it was produced by applying watershed segmentation and a variant of GALA agglomeration to the predictions made by an Ilastik [25]-trained voxel classifier.

We tested both GALA and CELIS using the same initial oversegmentations for the training and test regions. To compare the accuracy of the reconstructions, we computed two measures of segmentation consistency relative to the ground truth: Variation of Information [21] and Rand $F_1$ score, defined as the $F_1$ classification score over connectivity between all voxel pairs within the volumes; these are the primary metrics used in prior work. [28, 8, 22] The former has the advantage of weighing segments linearly in their size rather than quadratically.

Because any agglomeration method is ultimately limited by the quality of the initial oversegmentation, we also computed the accuracy of an *oracle* agglomeration policy that greedily optimizes the error metric directly. (Computing the true globally-optimal agglomeration under either metric is intractable.) This serves as an (approximate) upper bound that is useful for separating the error due to agglomeration from the error due to the initial oversegmentation.

## 6 Results

Figure 3 shows the Pareto optimal trade-offs between test set split and merge error of each method obtained by varying the choice of hyperparameters and agglomeration thresholds, as well as the Variation of Information and Rand $F_1$ scores obtained from the training set-optimal hyperparameters. CELIS consistently outperforms all other methods by a significant margin under both metrics. The large gap between the Oracle results and the best automated reconstruction indicates, however, that there is still large room for improvement in agglomeration.

While the evaluations are done on a single dataset, it is a single very *large* dataset; to verify that the improvement due to CELIS is broad and general (rather than localized to a very specific part of the image volume), we also evaluated accuracy independently on 18 non-overlapping $500^3$-voxel subvolumes evenly spaced within the test region. On all subvolumes CELIS outperformed the best existing method under both metrics, with a median reduction in Variation of Information error of 19% and in Rand $F_1$ error of 22%. This suggests that CELIS is improving accuracy in many parts of the volume that span significant variations in shape and image characteristics.

# 7 Discussion

We have introduced CELIS, a framework for modeling image segmentations using a learned energy function that specifically exploits the combinatorial nature of dense segmentation. We have described how this approach can be used to model the conditional energy of a segmentation given an image, and how the resulting model can be used to guide supervoxel agglomeration decisions. In our experiments on a challenging 3d microscopy reconstruction problem, CELIS improved volumetric reconstruction accuracy by 20% over the best existing method, and offered a strictly better trade-off between split and merge errors, by a wide margin, compared to existing methods.

The experimental results are unique in the scale of the evaluations: the 11-gigavoxel test region is 2–4 orders of magnitude larger than used for evaluation in prior work, and we believe this large scale of evaluation to be critically important; we have found evaluations on smaller volumes, containing only short neurite fragments, to be unreliable at predicting accuracy on larger volumes (where propagation of merge errors is a major challenge). While more computationally expensive than many prior methods, CELIS is nonetheless practical: we have successfully run CELIS on volumes approaching $\approx 1$ teravoxel in a matter of hours, albeit using many thousands of CPU cores.

In addition to advancing the state of the art in learning-based image segmentation, this work also has significant implications for the application area we have studied, connectomic reconstruction. The FIB-25 dataset reflects state-of-the-art techniques in sample preparation and imaging for large-scale neuron reconstruction, and in particular is highly representative of much larger datasets actively being collected (e.g. of a full adult fly brain). We expect, therefore, that the significant improvements in automated reconstruction accuracy made by CELIS on this dataset will directly translate to a corresponding decrease in human proof-reading effort required to reconstruct a given volume of tissue, and a corresponding increase in the total size of neural circuit that may reasonably be reconstructed.

Future work in several specific areas seems particularly fruitful:

- End-to-end training of the CELIS energy modeling pipeline, including the CNN model for computing the image feature representation and the aggregation of local energies at each position and scale. Because the existing pipeline is fully differentiable, it is directly amenable to end-to-end training.
- Integration of the CELIS energy model with discriminative training of a neural network-based agglomeration policy. Such a policy could depend on the *distribution* of local energy changes, rather than just the sum, as well as other per-object and per-action features proposed in prior work. [22, 3]
- Use of a CELIS energy model for fixing undersegmentation errors. While the energy minimization procedure proposed in this paper is based on a greedy local search limited to performing merges, the CELIS energy model is capable of evaluating arbitrary changes to the segmentation. Evaluation of candidate splits (based on a hierarchical initial segmentation or other heuristic criteria) would allow for the use of a potentially more robust simulated annealing energy minimization procedure capable of both splits and merges.

Several recent works [24, 32, 7, 6] have integrated deep neural networks into pairwise-potential conditional random field models. Similar to CELIS, these approaches combine deep learning with structured prediction, but differ from CELIS in several key ways:

- Through a restriction to models that can be factored into pairwise potentials, these approaches are able to use mean field and pseudomarginal approximations to perform efficient approximate inference. The CELIS energy model, in contrast, sacrifices factorization for the richer combinatorial modeling provided by the proposed 3-D shape descriptors.
- More generally, these prior CRF methods are focused on refining predictions (e.g. improving boundary localization/detail for semantic segmentation) made by a feed-forward neural network that are correct at a high level. In contrast, CELIS is designed to correct fundamental inaccuracy of the feed-forward convolutional neural network in critical cases of ambiguity, which is reflected in the much greater complexity of the structured model.

**Acknowledgments**

This material is based upon work supported by the National Science Foundation under Grant No. 1118055.

## Footnotes

[1]While prior work [30, 14, 2] has recognized the importance of combinatorial reasoning, the previously proposed global optimization methods allow local decisions to interact only in a very limited way.

[2]The BRIEF descriptor [5] is similarly defined as a binary descriptor based on a subset of the pairs of points within a patch, but each bit is based on the intensity difference, rather than connectivity, between each pair.

[3]For example, if most of the weight is on false merge examples, as would often occur without balancing, the model can simply learn to assign a score that increases with the number of 1 bits in the shape descriptor.

[4]GALA also supports multi-channel image features, potentially representing predicted probabilities of additional classes, such as mitochondria, but we did not make use of this functionality as we did not have training data for additional classes.

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
