[Supplementary Material]

Table 1: GALA and CELIS results on 18 non-overlapping $500^3$-voxel subvolumes within FIB-25 test region. Each subvolume is identified by the (x, y, z) coordinates of its start corner.

| Subvolume | GALA | | CELIS | |
|---|---|---|---|---|
| | VI | Rand $F_1$ | VI | Rand $F_1$ |
| (3056, 2228, 5006) | 0.710920 | 0.877058 | 0.639525 | 0.883413 |
| (3684, 2228, 5006) | 0.852962 | 0.807559 | 0.742159 | 0.847611 |
| (2428, 2856, 5006) | 0.550208 | 0.939376 | 0.543177 | 0.948212 |
| (3056, 2856, 5006) | 0.902916 | 0.763665 | 0.676395 | 0.801584 |
| (3056, 2228, 5634) | 0.825079 | 0.830070 | 0.674668 | 0.880949 |
| (3684, 2228, 5634) | 0.912993 | 0.843953 | 0.692192 | 0.868810 |
| (2428, 2856, 5634) | 0.806402 | 0.866520 | 0.731852 | 0.893787 |
| (3056, 2856, 5634) | 0.896207 | 0.882145 | 0.748106 | 0.903290 |
| (2428, 2228, 6262) | 0.724371 | 0.901122 | 0.579692 | 0.941264 |
| (3056, 2228, 6262) | 0.991092 | 0.848851 | 0.806581 | 0.897247 |
| (3684, 2228, 6262) | 0.971468 | 0.787515 | 0.747754 | 0.861740 |
| (2428, 2856, 6262) | 0.881123 | 0.869841 | 0.795054 | 0.897715 |
| (3056, 2856, 6262) | 1.113600 | 0.844898 | 0.877817 | 0.904204 |
| (2428, 2228, 6890) | 0.963757 | 0.902604 | 0.733394 | 0.953988 |
| (3056, 2228, 6890) | 0.983061 | 0.844851 | 0.870729 | 0.854900 |
| (3684, 2228, 6890) | 0.966499 | 0.883959 | 0.764519 | 0.910047 |
| (2428, 2856, 6890) | 1.380077 | 0.710782 | 0.869191 | 0.821951 |
| (3056, 2856, 6890) | 1.128732 | 0.713933 | 0.916494 | 0.846875 |

Table 2: Network architecture used for oversegmentation and image features.

| Layer | Input | Transform | Output | # parameters | Dropout ($p$) |
|---|---|---|---|---|---|
| 1 | $1 \times 35 \times 35 \times 9$ | $5 \times 5 \times 1$ convolution, ReLU | $64 \times 31 \times 31 \times 9$ | $64 \cdot (5^2 + 1)$ | 0.9 |
| 2 | $64 \times 31 \times 31 \times 9$ | $5 \times 5 \times 5$ convolution, ReLU | $64 \times 27 \times 27 \times 5$ | $64 \cdot (64 \cdot 5^3 + 1)$ | 0.9 |
| 3 | $64 \times 27 \times 27 \times 5$ | $2 \times 2 \times 1$ max pooling | $64 \times 14 \times 14 \times 5$ | | 0.9 |
| 4 | $64 \times 14 \times 14 \times 5$ | $5 \times 5 \times 5$ convolution, ReLU | $64 \times 10 \times 10 \times 1$ | $64 \cdot (64 \cdot 5^3 + 1)$ | 0.9 |
| 5 | $64 \times 10 \times 10 \times 1$ | $2 \times 2 \times 1$ max pooling | $64 \times 5 \times 5 \times 1$ | | 0.9 |
| 6 | $64 \times 5 \times 5 \times 1$ | Fully-connected ReLU | 512 | $512 \cdot (64 \cdot 5^2 + 1)$ | 0.5 |
| 7 | 512 | Fully-connected logistic | 3 | $3 \cdot (512 + 1)$ | |

(a) Small-scale                    (b) Large-scale

Figure 1: Connectivity region tiling. The connected components of the segmentation within each connectivity region $C$ (shown in distinct colors) are maintained independently. The yellow rectangle within each connectivity region indicates the bounds of $X_C^s$, the set of (type $s$) shape descriptor center positions computed using $C$, which is simply the set of center positions for which the shape descriptor bounding box is contained within $C$. The white rectangle (of size $B_s$) indicates the bounding box of the shape descriptor (necessarily contained within $C$).

Figure 2: Examples of cases where local boundary classification alone leads to false splits of neurites. A cross-section of the raw data is shown on the left; the correct segmentation (determined by careful human annotators) of the central neurite is overlayed on the right. Neuronal processes often narrow to nearly the limit of the image resolution, and when this is coupled with a loss of contrast, it appears to be impossible to determine the correct segmentation from local boundary information alone. These examples are from a Drosophila larval neuropil dataset [4] imaged using Focused Ion Beam Scanning Electron Microscopy (FIBSEM) [4].

Figure 3: Examples of cases where independent neurite shape modeling breaks down. At these synapse sites, the pre-synaptic and post-synaptic neurons each have characteristic shapes that are highly unlikely to occur independently but are jointly very likely. Due to the close contact between the two neurons, local boundary classification at these sites often results in false mergers, making correct shape modeling particularly critical. A cross-section of the raw data is shown on the left; the correct segmentation (determined by careful human annotators) is overlayed on the right. These examples are from a Drosophila larval neuropil dataset [4] imaged using Focused Ion Beam Scanning Electron Microscopy (FIBSEM) [4].

Figure 4: Distinction between local and global connectivity. In the cross-section of raw data on the left, there is clear evidence that the two points indicated within the yellow bounding box are separated by cell membrane. From the manual annotation overlaid on the right, it is clear, however, that they are nonetheless part of the same cell, highlighted in red. Thus, within a sufficiently local area the two points are disconnected, but globally they are connected. Distinguishing the connectivity of points at *multiple scales* is critical for accurate shape modeling. If connectivity is represented only globally, as in prior agglomeration work [1, 4], it may be impossible to reconcile strong local evidence of a cell boundary between two parts of the same sell in cases of self-contact, leading to poor learning and incorrect predictions for these cases. This example is from a Drosophila larval neuropil dataset [4] imaged using Focused Ion Beam Scanning Electron Microscopy (FIBSEM) [4].

# A  Local Connectivity

To reliably distinguish between local and global connectivity, we represent segmentations globally as an *undirected graph over voxels*. The vertices of this graph correspond to positions in $\mathbb{Z}^3$, and edges are typically limited to occur between *neighboring* voxel positions, for some definition of neighboring. We will define our neighborhood $\mathcal{N}(x)$ to be the von Neumann neighborhood (6-connectivity), though the Moore neighborhood or any other (symmetric) neighborhood could equally well be used.

The segments themselves are *implicitly* defined by the connected components of this graph, in contrast to a representation defined by an explicit labeling of voxels by the component to which they belong. The advantage of this representation is illustrated in Fig. 5.

# B  Local shape descriptors

Each connectivity region $C$ is a rectangular subset of the full volume.

**Definition 1.** Given a shape descriptor specification $s$ and connectivity region $C$, we denote by $X_C^s$ the set of (type $s$) shape descriptor center positions for which the descriptor bounding box is contained within $C$.

*Remark.* Note that $X_C^s$ is a rectangular region obtained by simply shrinking the rectangular region $C$ by $(B_s - 1)/2$ on all sides (recall that $B_s$ is the shape descriptor bounding box for type $s$ shape descriptors).

We wish to represent shape information at multiple scales, and to represent both the joint shape of nearby objects as well as the shape of individual objects. Therefore, rather than using a *single* shape descriptor specification $s$ and a single connectivity region tiling, we use a *set* of shape descriptor specifications $s$, each implicitly associated with a particular choice of connectivity region size $\bar{B}_s$ and stride $\text{stride}_s$ (specified by 3-D vectors of integers) that define a overlapped tiling of the full segmentation space.

**Definition 2.** We define $\mathcal{C}_s$ to be the set of connectivity regions obtained as regular overlapping tiles of size $\bar{B}_s$ and stride $\text{stride}_s$.

To ensure that the bounding box for a shape descriptor at a given position is contained in exactly one connectivity region, we constrain $\bar{B}_s$ and $\text{stride}_s$ as follows:

$$B_s \leq \bar{B}_s;$$
$$B_s = \bar{B}_s - \text{stride}_s + 1.$$

These constraints ensure that $\mathcal{C}_s$ *exactly partitions* the set of shape descriptor center positions, which allows us to make the following definition:

**Definition 3.** We denote by $\mathcal{C}_s(x)$ the single $C \in \mathcal{C}_s$ such that $x \in X_C^s$.

For convenience, we will also introduce some notation that applies to general undirected graphs that is relevant to our discussion:

**Definition 4** (Connected components). Given an undirected graph $G$, we denote by $\mathcal{K}(G)$ the partition of the vertex set of $G$ into connected components, and denote by $K(v; G)$ the connected component $G$ containing the vertex $v$.

**Definition 5** (Induced subgraph). Given an undirected graph $G$ and a subset $V'$ of its vertices, we denote by $G[V']$ the subgraph of $G$ induced by $V'$.

While globally we will represent a segmentation $S$ as a voxel graph, within a given connectivity region $C$ we are concerned only with the connected components $\mathcal{K}(S[C])$ in the subgraph of $S$ induced by $C$. Note that because the vertices of $S$ correspond to voxels, i.e. positions in $\mathbb{Z}^3$, $\mathcal{K}(S[C]) \subset 2^{\mathbb{Z}^3}$. Based on these definition, we can more precisely state how local shape descriptors are defined.

(a) Graph representation      (b) Component representation      (c) Component representation

Figure 5: Advantage of voxel graph representation. The top row shows a representation of a segmentation as either a voxel graph or a component labeling. The bottom row shows the effect of restricting the segmentation to a sub-region. Each square corresponds to a voxel. In the graph representation, a white line between two voxels indicates an edge, while a black line indicates the lack of an edge. In the component representation, each voxel is labeled by a component identifier (0 or 1). The different colors (red, blue, and grey) correspond to different connected components. The graph representation, shown on the left, correctly disconnects the two parts when restricted to the sub-region. The component labeling representation, shown in the middle, is unable to represent the presence of a boundary between the two parts, and therefore incorrectly results in a single connected component even when restricted to the sub-region. It is possible to emulate a voxel graph using a component representation by indicating boundaries with a 1-voxel wide background component, as shown on the right, but this tends to be cumbersome.

**Definition 6.** Given a full segmentation $S$, for each shape descriptor specification $s$, we define the $|s|$-bit local binary shape descriptor $r_s(x; S)$ at position $x$ by

$$r_s^{\{a,b\}}(x; S) := \mathbb{1}_{K(x+a; S[C]) = K(x+b; S[C])} \qquad \text{for } \{a, b\} \in s,$$

where $C = \mathcal{C}_s(x)$.

**Definition 7.** Given a segmentation $S$, we define the *component visibility set* $V_s(x; S) \subseteq \mathcal{K}(S[C])$ of a position $x$ to be the set of connected components at positions sampled by the shape descriptor $s$:

$$V_s(x; S) := \{ K(x+c; S[C]) \mid c \in \{a, b\} \in s \},$$

where $C = \mathcal{C}_s(x)$.

**Lemma 1.** *Let a shape descriptor specification $s$, a position $x$, and segmentations $S$ and $S'$ be given. Let $C = \mathcal{C}_s(x)$. If $V_s(x; S) = V_s(x; S')$ (in particular if $\mathcal{K}(S[C]) = \mathcal{K}(S'[C])$), then $r_s(x; S) = r_s(x; S')$. Furthermore, in the case that $s$ is center-based, then if $K(x; S[C]) = K(x; S'[C])$, then $r_s(x; S) = r_s(x; S')$.*

*Proof.* The first statement follows directly from Definition 6.

To prove the second statement, suppose that $s$ is center-based. Note that for all $\{a, b\} \in s$, $\{a, b\} = \{\vec{0}, c\}$ for $c \in \{a, b\}$. Thus, we have

$$
\begin{aligned}
r_s^{\{a,b\}}(x; S) &= r_s^{\{\vec{0},c\}}(x; S) \\
&= \mathbb{1}_{K(x; S[C]) = K(x+c; S[C])} \\
&= \mathbb{1}_{(x+c) \in K(x; S[C])} \qquad \text{for } \{\vec{0}, c\} = \{a, b\} \in s.
\end{aligned}
$$

The result follows. $\qquad \square$

*Remark.* For general shape descriptor specifications $s$, $r_s(x; S)$ depends on $S$ only by way of the subset of $\mathcal{K}(S[\mathcal{C}_s(x)])$ that are sampled, and for center-based shape descriptor specifications, $r_s(x; S)$ depends on $S$ only by way of $K(x; S[\mathcal{C}_s(x)])$, the single component in $S[C]$ that contains $x$.

## C    Efficient energy minimization

Naïvely, computing the energy for just a single segmentation requires computing shape descriptors and then evaluating the energy model at every voxel position with the volume; a small volume may have tens or hundreds of millions of voxels. At each stage of the agglomeration, there may be thousands, or tens of thousands, of potential next agglomeration steps, each of which results in a unique segmentation. In order to choose the best next step, we must know the energy of all of these potential next segmentations. The computational cost to perform these computations *directly* would be tremendous.

We will discuss several computational tricks that allow us to efficiently compute these energy terms *incrementally*. Because the cost of evaluating the local energy model for a single shape descriptor is many times more expensive than computing the shape descriptor, we structure our computation such that we only recompute a local energy term if the shape descriptor on which it depends has changed. This ensures that the total cost of evaluating the local energy terms is minimized, but even computing just the shape descriptors at each position within the volume for each potential agglomeration action at each step would still be prohibitively expensive. We therefore rely on geometric and region graph information to prune out the vast majority of this computation as well. Collectively, these tricks reduce the computational cost by several orders of magnitude; the effectiveness of these techniques is ultimately data-dependent, however.

## C.1 Action representation

Recall that each agglomeration action $e$ corresponds to a set of additional voxel edges to be added to the current segmentation state $S^t$. While in principle agglomeration could be defined with respect to arbitrary sets of voxel edges, we will carefully choose the set of actions to be considered in order to preserve the distinction between local and global connectivity while also allowing for a computationally-efficient implementation.

We will define actions in terms of adjacent supervoxels $K, K' \in \mathcal{K}(S^0)$ in the *initial* segmentation:

**Definition 8.** For any two distinct connected components $K, K' \in \mathcal{K}(S^0)$, let

$$e_{K,K'} := \{\{x, x'\} \,|\, x' \in \mathcal{N}(x) \wedge (x, x') \in K \times K'\}.$$

*Remark.* If $K$ and $K'$ are not adjacent, then $e_{K,K'} = \varnothing$.

Note that we represent edges in the undirected voxel graph simply as two-element sets of voxel positions.

**Definition 9.** We define the supervoxel merge action set

$$A_S := \{e_{K,K'} \neq \varnothing \,|\, K, K' \in \mathcal{K}(S) \wedge K \neq K'\}.$$

We will use $A^0 := A_{S_0}$ as our set of actions for agglomeration. Note that each action corresponds to a *set* of voxel graph edges. At each step $t$ of agglomeration, we choose an action $e^t \in A^t$. The set of remaining actions $A^t$ after step $t$ is simply the subset of actions in $A$ that have not yet been performed, i.e. $A^{t+1} = A^t - \{e^t\}$. The segmentation state $S^{t+1} := S^t + e^t$.

**Definition 10.** If $e$ is a set of edges and $C$ is a set of vertices, we denote by $e[C]$ the restriction of $e$ to vertices in $C$, i.e. the subset of edges in $e$ that are incident to two vertices in $C$. If $S$ is a graph, we define $e[S] := e[\text{vertices}(S)]$ to be the restriction of $e$ to vertices in $S$.

Given a graph $S$ and a partition $T$ of $\text{vertices}(S)$, we denote by $G/T$ the contraction of $G$ by $T$.

**Definition 11.** A set $e$ of voxel edges is said to be a *supervoxel merge* in a voxel graph $S$ of components $K, K' \in \mathcal{K}(S)$ if $e[S]$ is a non-empty set of edges between components $K$ and $K'$, or equivalently, that every edge in $e[S]$ corresponds to the edge $\{K, K'\}$ in $S/\mathcal{K}(S)$.

**Definition 12.** A set $e$ of voxel edges is said to be a *redundant merge* in a voxel graph $S$, corresponding to the component $K \in \mathcal{K}(S)$, if $e[S]$ is a non-empty set of edges within component $K$, i.e. $\{\{K(a; S), K(b; S)\} \,|\, \{a, b\} \in e\} = \{\{K\}\}$, or equivalently, that every edge in $e$ corresponds to a self edge $\{K\}$ in $S/\mathcal{K}(S)$.

**Lemma 2.** *If $e$ is a redundant merge in $S$, then $\mathcal{K}(S + e) = \mathcal{K}(S)$.*

*Proof.* This follows from the fact that adding an edge between two vertices already part of the same connected component does not change set of connected components. $\square$

**Definition 13.** Let $e, e'$ be supervoxel merges in $S$. We say that $e$ is *incident to a connected component* $K \in \mathcal{K}(S)$ in $S$ if every edge in $e$ is incident to a voxel in $K$, i.e. $e$ is incident to $K$ in $S/\mathcal{K}(S)$. We say that $e$ is incident to $e'$ in $S$ if there exists $K \in components S$ to which both $e$ and $e'$ are incident, i.e. $e$ is incident to $e'$ in $S/\mathcal{K}(S)$.

**Lemma 3.** *If $e$ is a supervoxel merge in $S$ and $S$ is a spanning subgraph of $S'$, then $e$ is a supervoxel merge or redundant merge in $S'$. If $e$ is a redundant merge in $S$, then $e$ is a redundant merge in $S'$.*

*Proof.* Suppose $e$ is a supervoxel merge in $S$, corresponding to $K, K' \in \mathcal{K}(S)$. There must exist a components $J, J' \in \mathcal{K}(S')$ with $K \subseteq J$ and $K' \subseteq J'$. If $J = J'$, then $e$ is a redundant merge in $S'$; otherwise $e$ is a supervoxel merge of $\{J, J'\}$.

Suppose $e$ is a redundant merge in $S$ corresponding to $K \in \mathcal{K}(S)$. There must exist a component $J \in \mathcal{K}(S')$ with $K \subseteq J$. Hence, $e$ is a redundant merge in $S'$ corresponding to $J$. $\square$

*Remark.* $S$ is necessarily a spanning subgraph of $S + e$ for any merge action $e$.

**Lemma 4.** *At all steps $t$, all $e \in A$ are either supervoxel merges or redundant merges in $S^t$.*

*Proof.* This follows from Lemma 3 and the fact that all $e \in A$ are supervoxel merges in $S^0$. $\square$

The consequence of this lemma is that globally each merge action corresponds to a pair of connected components. Within the induced subgraph $S^t[C]$ of $S^t$ restricted to a given connectivity region $C$, however, this lemma does not necessarily hold, even for $S^0[C]$, because a connected component of $S^0$ may correspond to more than one connected component of $S^0[C]$. For computational reasons that will be made apparent in Appendix C.2, we would like to ensure that it *does* hold, so that each merge action also corresponds to a pair of connected components within each connectivity region $C$ (or is redundant within $C$).

To do this, we will assume that each connected component of $S^0$ is a clique. Our assumption sacrifices any distinction between local and global connectivity within the original supervoxels of $S^0$, but this is a small sacrifice given that they are expected to be small.

**Lemma 5.** *Given a connectivity region $C$, if $e$ is a supervoxel merge in $S^0$ and $e[C]$ is non-empty, then $e$ is either a supervoxel merge or a redundant merge in $S^t[C]$ for all $t$.*

*Proof.* Suppose $e$ is a supervoxel merge in $S^0$ of components $K_1, K_2 \in \mathcal{K}(S^0)$. For all $\{a, b\} \in e[C]$, without loss of generality we can assume $a \in K_1$ and $b \in K_2$. By our assumption that $K_1$ and $K_2$ are cliques in $S^0$, $K_1 \cap C, K_2 \cap C \in \mathcal{K}(S^0[C])$. By the definition of $e[C]$, we have $a \in K_1 \cap C$ and $b \in K_2 \cap C$. Hence, $e$ is a supervoxel merge in $S^0[C]$. The result follows from Lemma 3. $\square$

## C.2    $\Delta$ representation

Recall that we defined the forward discrete derivative of $f$ with respect to $S$ by:

$$\Delta_S^e f(S) := f(S + e) - f(S).$$

We also define the second discrete derivative:

$$\Delta_S^{e,e'} f(S) := \Delta_S^e \Delta_S^{e'} f(S).$$

To efficiently implement a local search over agglomerations, at each step $t$ of agglomeration, for each possible next agglomeration action $e$, we maintain the discrete derivative $\Delta_{S^t}^e E(S^t; I)$, where $S^t$ denotes the current segmentation at step $t$. Although our energy model is defined without any reference to supervoxels or merges, we prove a number of key properties that enable us to very efficiently compute and update these discrete derivative terms.

To maintain $\Delta_{S^t}^e E(S^t; I)$, conceptually we must initially compute $\Delta_{S^0}^e E_s(x; S^0; I)$ for each position $x$ and action $e$, and then at each subsequent step $t$, agglomeration action $a^t$ is taken and we update

$$
\begin{aligned}
\Delta_{S^{t+1}}^e & E(S^{t+1}; I) \\
&= \Delta_{S^t}^e E(S^t; I) \\
&\quad + \sum_s \sum_x \Delta_{S^t}^{e,e^t} E_s(x; S^t; I) \qquad\qquad \text{for all } e \in A^{t+1}.
\end{aligned}
$$

**Theorem 1** (Descriptor-based pruning)**.** *Let a position $x$ and image $I$ be given. Let $\bar{r}(S') := r_s(x; S')$ and $\bar{E}(S') := E_s(x; S'; I)$. Given a segmentation $S$, and merge $e$, if $\bar{r}(S) = \bar{r}(S + e)$, then $\Delta_S^e \bar{E}(S) = 0$. Furthermore, for any merge $e'$,*

$$
\begin{aligned}
d := \Delta_S^{e,e'} \bar{E}(S) = \\
&+\bar{E}(S) \qquad\quad -\bar{E}(S + e) \\
&-\bar{E}(S + e') \quad +\bar{E}(S + e' + e),
\end{aligned}
$$

*where some or all of the 4 terms can be canceled based on whether $\bar{r}(S) = \bar{r}(S + e)$, $\bar{r}(S) = \bar{r}(S + e')$, $\bar{r}(S + e) = \bar{r}(S + e' + e)$, and/or $\bar{r}(S + e') = \bar{r}(S + e' + e)$. In particular,*

$$\bar{r}(S) = \bar{r}(S + e) \ \wedge \ \bar{r}(S + e') = \bar{r}(S + e' + e)$$
$$\implies d = 0;$$
$$\bar{r}(S) \neq \bar{r}(S + e) \ \wedge \ \bar{r}(S + e') = \bar{r}(S + e' + e)$$
$$\implies d = \bar{E}(S) - \bar{E}(S + e);$$
$$\bar{r}(S) = \bar{r}(S + e) \ \wedge \ \bar{r}(S + e') \neq \bar{r}(S + e' + e)$$
$$\implies d = \bar{E}(S + e' + e) - \bar{E}(S + e').$$

*By symmetry of the theorem with respect to $e$ and $e'$ we also have:*

$$\bar{r}(S) = \bar{r}(S + e') \ \wedge \ \bar{r}(S + e) = \bar{r}(S + e' + e)$$
$$\implies d = 0;$$
$$\bar{r}(S) = \bar{r}(S + e') \ \wedge \ \bar{r}(S + e) \neq \bar{r}(S + e' + e)$$
$$\implies d = \bar{E}(S + e' + e) - \bar{E}(S + e);$$
$$\bar{r}(S) \neq \bar{r}(S + e') \ \wedge \ \bar{r}(S + e) = \bar{r}(S + e' + e)$$
$$\implies d = \bar{E}(S) - \bar{E}(S + e').$$

*Proof.* For the first statement, if $\bar{r}(S) = \bar{r}(S + e)$, we have

$$\bar{E}(S) = \hat{E}_s\left(\bar{r}(S); \phi(x; I)\right)$$
$$= \hat{E}_s\left(\bar{r}(S + e); \phi(x; I)\right)$$
$$= \bar{E}(S + e).$$

The result follows.

The second statement is a straightforward result of the same cancellation principle. $\qquad\square$

*Remark.* This theorem allows us to skip a large fraction of evaluations of the local energy model, which is in general significantly more expensive than just computing the shape descriptors (which must still be done in order to check the conditions of this theorem). If a packed bitvector representation is used, the cost of the descriptor comparisons is negligible.

## C.3   Connectivity region-based pruning

Recall that for every merge action $e$ exactly one of the following is true:

1. $e$ is a supervoxel merge in $S^t[C]$;

2. $e$ is a redundant merge in $S^t[C]$;

3. $e[C] = \varnothing$.

**Definition 14.** For each connectivity region $C$, we define the *active action set* $A^t[C] \subseteq A^t$ to be the subset of actions at step $t$ that are supervoxel merges in $S^t[C]$.

**Lemma 6.** *Given a connectivity region $C$, if $e \notin A^t[C]$, then $e \notin A^{t'}[C]$ for all $t' > t$.*

*Proof.* Suppose $e \notin A^t[C]$. Then either $e[C] = \varnothing$ or $e$ is a redundant merge in $S^t[C]$. If $e[C] = \varnothing$, then $e \notin A^{t'}[C]$ for any $t'$. Alternatively, if $e$ is a redundant merge in $S^t[C]$, then since $S^t[C]$ is a spanning subgraph of $S^{t'}[C]$, by Lemma 3 $e$ is a redundant merge in $S^{t'}[C]$. $\qquad\square$

**Theorem 2** (Connectivity region-based pruning). *Given a position $x$, time step $t$, and merge $e \in A^t$, let $C = \mathcal{C}_s(x)$ and let $A' = A^t[C]$. If $e \notin A'$, then $\Delta^e_{S^t} E_s(x; S^t; I) = 0$. Furthermore, if $\{e, e'\} \not\subseteq A'$, then $\Delta^{e,e'}_{S^t} E_s(x; S^t; I) = 0$.*

*Proof.* We will begin by proving the first statement. Suppose $e \notin A'$. By definition of $A'$, it follows that $e$ is a redundant edge in $S^t[C]$, i.e. $\mathcal{K}(S^t[C]) = \mathcal{K}((S^t + e)[C])$. By Lemma 1, we have $r_s(x; S^t) = r_s(x; S^t + e) = r$. The result follows from the first part of Theorem 1.

Next we will consider the second statement. Since $\Delta^{e,e'}_{S^t} E_s(x; S^t; I) = \Delta^{e',e}_{S^t} E_s(x; S^t; I)$, the second statement is symmetric with respect to $e$ and $e'$. It is sufficient, therefore to again consider the case that $e \notin A'$. By the first statement, $\Delta^e_{S^t} E_s(x; S^t; I) = 0$. Since $S^t$ is a spanning subgraph of $S^t + e'$, it is likewise the case that $e$ is a redundant merge in $(S^t + e')[C]$, which implies that $r_s(x; S^t + e') = r_s(x; S^t + e' + e)$. The result follows from the second part of Theorem 1. $\qquad\square$

*Remark.* Because each action is typically active in only a tiny fraction of the connectivity regions, this theorem allows us to dramatically limit our computation.

## C.4 Graph-based pruning

**Lemma 7.** *Let a segmentation $S$ and a supervoxel merge $e$ in $S$ be given. Let $K \in \mathcal{K}(S)$ be a connected component of $S$. If $e$ is not incident in $S$ to $K$, then $K \in \mathcal{K}(S + e)$, i.e. merging $e$ in $S$ does not affect $K$.*

*Proof.* This follows from the fact that by definition of incidence of a supervoxel merge, no edge in $e$ is incident to any voxel in $K$. $\qquad\square$

**Theorem 3** (Graph-based pruning). *Suppose $s$ defines a center-based descriptor. Let a segmentation $S$, position $x$, and supervoxel merges $e$ and $e'$ in $S$ be given. Let $C = \mathcal{C}_s(x)$. If $e$ is not incident in $S[C]$ to $K(x; S[C])$, then $r_s(x; S) = r_s(x; S + e)$ and $\Delta^e_S E_s(x; S; I) = 0$. Furthermore, if $e$ is not incident in $(S + e')[C]$ to $K(x; (S + e')[C])$, or $e'$ is not incident to $e$ in $S[C]$, then $\Delta^{e,e'}_S E_s(x; S; I) = 0$.*

*Proof.* We will being by proving the first statement. Suppose $e$ is not incident in $S[C]$ to $K := K(x; S[C])$. By Lemma 7, we have $K = K(x; (S + e)[C])$. By Lemma 1 this implies that $r_s(x; S) = r_s(x; S + e)$. The result follows from the first part of Theorem 1.

Next we will consider the second statement. Note that the condition that $e$ is incident in $(S + e')[C]$ to $K(x; (S + e')[C])$ is equivalent to the condition that $e$ is incident in $S[C]$ to $K := K(x; S[C])$, or $e'$ is a supervoxel merge of $K$ and $K'$ in $S[C]$ (i.e. incident to $e$ in $S[C]$) and $e$ is incident to $K'$ in $S[C]$.

There are two cases to consider: suppose $e$ is not incident in $(S + e')[C]$ to $K(x; (S + e')[C])$. Then since $S$ is a spanning subgraph of $S + e'$, it follows that $e$ is also not incident in $S[C]$ to $K(x; S[C])$. The result follows from applying the first statement of the theorem to both $S$ and $S + e'$ and then using Theorem 1.

Alternatively, suppose $e'$ is not incident to $e$ in $S[C]$. This implies that $K(x; S[C])$ is incident to at most one of $\{e, e'\}$ in $S[C]$. By the symmetry of the theorem with respect to $e$ and $e'$, we will assume without loss of generality that $e$ is not incident to $K(x; S[C])$ in $S[C]$. By our note above, we can infer that the condition for our first case, that $e$ is not incident to $K(x; (S + e')[C])$ in $(S + e')[C]$, holds. $\qquad\square$

*Remark.* This theorem demonstrates that for center-based descriptors, we can significantly limit computation based on the agglomeration graph structure. The cost of maintaining the incidence information is negligible.

## C.5 Visibility-based pruning

**Lemma 8.** *Let a position $x$, segmentation $S$ and supervoxel merge $e$ of components $K_1$ and $K_2$ in $S[C]$, where $C = \mathcal{C}_s(x)$, be given. If $e$ is incident in $S[C]$ to at most one component in $V_s(x; S)$, then $r_s(x; S) = r_s(x; S + e)$.*

*Proof.* There are two cases to consider. If $e$ is not incident in $S[C]$ to any component in $V_s(x; S)$, then by Lemma 7, $V_s(x; S) = V_s(x; S + e)$. The result follows from Lemma 1. If $e$ is incident in $S[C]$ to exactly one component $K_1 \in V_s(x; S)$, then $V_s(x; S + e') = V_s(x; S) + \{K'_1 \cup K'_2\} - \{K_1\}$, i.e. merging $e'$ in $S$ adds additional voxels (not part of any visible component) to one visible component. Since these additional voxels are, by definition, not sampled by the shape descriptor, it follows that $r_s(x; S) = r_s(x; S + e)$. $\quad\square$

**Theorem 4** (Visibility-based pruning). *Given a position $x$, and segmentation $S$, let $C = \mathcal{C}_s(x)$ Let $e'$ be a supervoxel merge of components $K_1$ and $K_2$ in $S[C]$. If $e'$ is not incident in $S[C]$ to any component $K_1 \in V_s(x; S)$, then $\Delta_S^e E_s(x; S; I) = 0$, and $\Delta_S^{e,e'} E_s(x; S; I) = 0$ for all supervoxel merges $e$ in $S[C]$. If $e'$ is incident in $S[C]$ to exactly one component $K_1 \in V_s(x; S)$, then for all supervoxel merges $e$ of $K'_1, K'_2$ in $S[C]$ not incident to $K_2$ in $S[C]$, i.e. $K_2 \notin \{K'_1, K'_2\}$, $\Delta_S^{e,e'} E_s(x; S; I) = 0$.*

*Proof.* To prove the first statement, suppose $e'$ is not incident in $S[C]$ to any component in $V_s(x; S)$. For any supervoxel merge $e$ in $S[C]$, it must be the case that $e'$ is incident in $(S + e)[C]$ to at most one component in $V_s(x; S + e)$. By applying Lemma 8 to both $S[C]$ and $S[C + e]$, we have $r_s(x; S) = r_s(x; S + e')$ and $r_s(x; S + e) = r_s(x; S + e + e')$. The result follows from Theorem 1.

To prove the second statement, suppose $e'$ is incident in $S[C]$ to exactly one component $K_1 \in V_s(x; S)$. As for the first statement, by Lemma 8 we have $r_s(x; S) = r_s(x; S + e')$. Let $e$ be a supervoxel merge of $K'_1, K'_2$ in $S[C]$ not incident to $K_2$ in $S[C]$. If $e$ is incident to $K_1$, then $e'$ is incident in $(S + e)[C]$ to exactly one component $(K'_1 + K'_2) \supseteq K_1$. If $e$ is not incident to $K_1$, then $e'$ is incident in $(S + e)[C]$ to exactly the one component $K_1$. Therefore, by Lemma 8 we have $r_s(x; S + e) = r_s(x; S + e + e')$ and the result follows from Theorem 1. $\quad\square$

Determining whether a given component $K \in S[C]$ is a member of the *exact* visibility set $V_s(x; S)$ for all positions $x \in X_C^s$ is computationally expensive, i.e. $\Theta(|X_C^s| \cdot |s|)$. However, to satisfy the conditions of Theorem 4, it is sufficient to check membership in any *superset* of the visibility set; this restricts the conditions under which pruning is done, but we can choose a superset in which membership can be checked much more efficiently.

**Definition 15.** For $d$-dimensional vectors $\vec{a}, \vec{b} \in \mathbb{Z}^d$, we denote by $R_{\vec{a}}^{\vec{b}}$ the hyperrectangle

$$R_{\vec{a}}^{\vec{b}} := \{\vec{x} \in \mathbb{Z}^d \mid \vec{a} \le \vec{x} < \vec{b}\}.$$

**Definition 16.** Given a segmentation $S$, we define the *approximate component visibility set* $\hat{V}_s(x; S) \subseteq \mathcal{K}(S[C])$ of a position $x$ to be the set of connected components at positions within a bounding box of size $B_s$ centered at $x$:

$$\hat{V}_s(x; S) := \left\{ K(x + c; S[C]) \,\middle|\, c \in R_{-(B_s - \vec{1})/2}^{(B_s - \vec{1})/2} \right\},$$

where $C = \mathcal{C}_s(x)$.

**Lemma 9.** *Given a segmentation $S$, $\hat{V}_s(x; S) \subseteq V_s(x; S)$.*

*Proof.* This follows from the fact that $\{a, b\} \subset R_{-(B_s - \vec{1})/2}^{(B_s - \vec{1})/2}$ for all $\{a, b\} \in s$. $\quad\square$

**Definition 17.** For two coordinate vectors $a$ and $b$, $a \odot b$ denotes the element-wise product.

For a given component $K \in S[C]$, by first computing a summed area table [3], we can efficiently determine whether $K \in \hat{V}_s(x; S[C])$ for all positions $x \in X_C^s$, as described in Algorithm 1. The computational cost is $\Theta(|C|)$. To check the conditions of Theorem 4 for a given supervoxel merge $e'$ of $K_1, K_2 \in S[C]$, we simply apply Algorithm 1 to both $K_1$ and $K_2$. Alternatively, to check only the (more limited) first condition that $\{K_1, K_2\} \cap V_s(x; S) = \varnothing$, then it is sufficient to apply Algorithm 1 just once to $K_1 \cup K_2$.

At agglomeration steps $t > 0$, we can apply Theorem 4 with $e' = a^{t-1}$ and $e \in A^t[C]$ in order to limit the set of positions $x$ and edges $e$ for which the change in local energy $\Delta_{S^t}^{e, e^t} E_s(x; S^t; I)$ must be computed.

**Algorithm 1** Optimized membership test for approximate component visibility sets.

---

**Require:** $(G, +)$ is a commutative group with identity $0_G$.

1: **function** COMPUTESUMMEDAREATABLE($A\colon R_a^b \to G$, $R_a^b$)
2:     **Declare** array $T\colon R_a^{b+\vec{1}} \to G$
3:     **for** $x \in R_a^{b+\vec{1}} : \|x - a\|_0 < d$ **do**
4:         $T(x) \leftarrow 0_G$
5:     **end for**
6:     **for** $x \in R_{a+\vec{1}}^b$ **do**               ▷ Iteration over $x$ must respect the usual partial ordering on $\mathbb{Z}^d$.
7:         $T(x) \leftarrow A(x - \vec{1}) + \displaystyle\sum_{z \in \{0,1\}^d - \{\vec{0}\}} (-1)^{1 + \|z\|_1} \cdot T(x - z)$
8:     **end for**
9:     **return** $T$
10: **end function**
11: **function** SUMMEDAREATABLELOOKUP($T\colon R_a^{b+\vec{1}} \to G$, $R_{a'}^{b'} \subseteq R_a^b$)
12:     **return** $\displaystyle\sum_{z \in \{0,1\}^d} (-1)^{\|z\|_1} \cdot T(b + (a - b) \odot z)$
13: **end function**
14: **function** COMPUTEPOSITIONSWITHVISIBILITY($s$, $S$, $C$, $K \in S[C]$)
15:     **Define** $A(x) \coloneqq \mathbb{1}_{K = K(x; S[C])}$
16:     $T \leftarrow$ COMPUTESUMMEDAREATABLE($A, C$)
17:     $X \leftarrow \varnothing$
18:     **for** $x \in X_C^s$ **do**
19:         **if** SUMMEDAREATABLELOOKUP($T, R_{x-(B_s-\vec{1})/2}^{x+(B_s-\vec{1})/2}$) $> 0$ **then**
20:             $X \leftarrow X \cup \{x\}$
21:         **end if**
22:     **end for**
23:     **return** $X$
24: **end function**

---

In principle, we could apply Theorem 4 to all candidate actions $e' \in A^t[C]$ at a given agglomeration step $t$, but this would require computing separate summed area tables for all components $K \in \mathcal{K}(S^t[C])$ incident to a candidate action, which would involve considerable overhead. Therefore in practice the theorem is only applicable for $t > 0$.

## C.6 Zone-based pruning

In the case of a pairwise shape descriptor specification $s$, we cannot apply Theorem 3, and consequently based only on Theorem 2, for each position $x$ we must compute shape descriptors for all actions $e \in A^t[\mathcal{C}_s(x)]$. Theorem 4 primarily allows us to prune positions $x$ but not actions $e$, and is not applicable at the initial state $t = 0$.

At $t = 0$, the number of positions that must be considered within a given connectivity region $C$ is exactly $|X_C^s|$; at later steps $t > 0$ the number of positions may be reduced due to Theorem 4 but nonetheless tends to grow linearly with $|X_C^s|$. The size of the active set $A^t[C]$ tends to grow superlinearly in $|C|$. Hence, the computational cost of shape descriptor computation based only on the pruning theorems we've introduced thus far grows superquadratically in $|C|$.

To mitigate this effect, we could of course simply ensure that connectivity regions are very small. A larger number of small connectivity regions does, however, introduce additional overhead, as explained in Appendix D, and therefore may actually increase the computational cost. Furthermore, reducing the connectivity region size also affects the extent to which shape descriptors reflect local or global connectivity, and we would like to be able to choose that independently of computational concerns.

We therefore introduce a subdivision of connectivity regions into *zones*.

**Definition 18.** For each connectivity region $C \in \mathcal{C}_s$, the *zone set* $\mathcal{Z}_{s,C}$ is a partition of $X_C^s$.

We can extend our definition of component visibility sets, previously defined only for individual positions in Definition 7, to sets of positions:

**Definition 19.** The *component visibility set* $W_s(Z; S)$ for a zone $Z$ is defined by

$$W_s(Z; S) \coloneqq \cup_{x \in Z} V_s(x; S).$$

**Definition 20.** The *zone visibility set* $W_s^{-1}(K; C)$ is the set of zones whose component visibility set contains $K$:

$$W_s^{-1}(K; C) \coloneqq \{Z \in \mathcal{Z}_{s,C} \,|\, K \in W_s(Z; S)\},$$

where $S$ is some segmentation for which $K \in \mathcal{K}(S[C])$.

*Remark.* The zone visibility set does not depend on the segmentation $S$ beyond the fact that $K \in \mathcal{K}(S[C])$. By definition, a merge that does not affect a connected component $K'$ does not affect its zone visibility set $W_s^{-1}(K'; C)$.

**Theorem 5.** *Given a supervoxel merge $e$ of $K_1, K_2$ in $S[C]$, merging $e$ in $S$ has the effect of merging the zone visibility sets of $K_1$ and $K_2$:*

$$W_s^{-1}(K_1 \cup K_2; C) = W_s^{-1}(K_1; C) \cup W_s^{-1}(K_2; C).$$

*Proof.* To show that the $W_s^{-1}(K_1; C) \cup W_s^{-1}(K_2; C)$ contains $W_s^{-1}(K_1 \cup K_2; C)$, let $Z \in W_s^{-1}(K_1 \cup K_2; C)$ be given. Then $\exists x \in Z, c \in \{a, b\} \in s$ such that $K(x + c; S'[C]) = (K_1 \cup K_2)$, where $S'$ is the segmentation that results from the merge of $K_1$ and $K_2$ in $S$. Hence, $K(x + c; S[C]) \subset \{K_1, K_2\}$, and it follows that $Z \in W_s^{-1}(K_1; C) \cup W_s^{-1}(K_2; C)$.

To show that $W_s^{-1}(K_1 \cup K_2; C)$ contains $W_s^{-1}(K_1; C) \cup W_s^{-1}(K_2; C)$, let $Z \in W_s^{-1}(K_1; C)$ be given. Then $\exists x \in Z, c \in \{a, b\} \in s$ such that $K(x + c; S[C]) = K_1$. It follows that $K(x + c; S'[C]) = (K_1 \cup K_2)$, and therefore $Z \in W_s^{-1}(K_1 \cup K_2; C)$. $\square$

**Definition 21.** A supervoxel merge $e$ of $K_1, K_2$ in $S[C]$ is said to be *active in zone $Z$ of $S[C]$* if $Z \in W_s^{-1}(K_1; C) \cap W_s^{-1}(K_2; C)$.

**Definition 22.** We denote by $A_Z^t[C]$ the *active action set of zone $Z$* of connectivity region $C$ at time $t$, the set of actions $e$ in $A^t[C]$ that are active in zone $Z$ of $S^t[C]$.

**Theorem 6.** *If a supervoxel merge $e$ in $S[C]$ is not active in zone $Z$, then for all positions $x \in Z$ we have $r_s(x; S) = r_s(x; S + e)$, $\Delta_S^e E_s(x; S; I) = 0$. Furthermore, given a supervoxel merge $e'$ in $S[C]$, if a supervoxel merge $e$ in $(S + e')[C]$ is not active in zone $Z$, then for all positions $x \in Z$, $\Delta_S^{e; e'} E_s(x; S; I) = 0$.*

*Proof.* To prove the first statement, suppose the supervoxel merge $e$ in $S[C]$ of components $K, K'$ is not active in zone $Z$, and $x \in Z$. Then by Definition 21, $\{K, K'\} \not\subseteq V_s(x; S)$. Hence, by Lemma 8 $r_s(x; S) = r_s(x; S + e)$, and the result follows from Theorem 1.

To prove the second statement, suppose the supervoxel merge $e$ of components $K_1, K_2$ in $(S + e')[C]$ is not active in zone $Z$ of $(S + e')[C]$. By the first statement of the theorem, this implies that $r_s(x; S + e') = r_s(x; S + e' + e)$ for all $x \in Z$. It remains to be shown that $e$ is also not active in zone $Z$ of $S[C]$. By Definition 21,

$$Z \notin W_s^{-1}(K_1; C) \cap W_s^{-1}(K_2; C). \tag{1}$$

There are two cases to consider:

1. If $e$ is not incident to $e'$ in $S[C]$, then $\{K_1, K_2\} \subset \mathcal{K}(S[C])$ and it follows from Eq. (1) and Definition 21 that $e$ is also not active in zone $Z$ of $S[C]$.

2. Alternatively, if $e$ is incident to $e'$ in $S[C]$, then without loss of generality we can assume that $K_2 \subset \mathcal{K}(S[C])$ and $K_1 = K_1' \cup K_2'$, where $e'$ is a supervoxel merge of $K_1', K_2'$ in $S[C]$, and $e$ is a supervoxel merge of $K_1', K_2$ in $S[C]$. Then by Theorem 5,

$$
\begin{aligned}
W_s^{-1}(K_1; C) &= W_s^{-1}(K_1' \cup K_2'; C) \\
&= W_s^{-1}(K_1'; C) \cup W_s^{-1}(K_1'; C).
\end{aligned}
$$

It follows that $Z \notin W_s^{-1}(K_1'; C) \cap W_s^{-1}(K_2; C)$, which by Definition 21 implies that $e$ is not active in zone $Z$ of $S[C]$. By the first statement of the theorem, we have $r_s(x; S) = r_s(x; S + e)$ for all $x \in Z$. The result follows from Theorem 1. $\square$

If we ensure that the total number of zones, $|\mathcal{Z}_{s,C}|$, is limited to a small constant, e.g. 64, then we can efficiently represent the zone visibility set $W_s^{-1}(K; C)$ for each component $K$ as a bit vector. Maintaining these visibility sets over the course of agglomeration, per Theorem 5, requires only bitwise disjunction operations; determining whether a supervoxel merge $e$ is active in a zone $Z$, per Definition 21, requires only bitwise conjunction.

Based on Theorem 6, the cost of computing all unpruned shape descriptors within a connectivity region $C$ can be formulated as

$$\sum_{Z \in \mathcal{Z}_{s,C}} \left[ (|Z| + \alpha) \cdot |A_Z^t[C]| \right] + \beta(|\mathcal{Z}_{s,C}|),$$

where $\alpha$ represents the overhead per action active in a zone, and $\beta$ is a non-decreasing function that specifies an additional overhead for a given number of zones; $\alpha$ and $\beta$ may represent either computational or memory costs.

Minimizing this cost exactly is in general a hard integer programming problem. We find a locally-optimal solution using an approach that mirrors our approach for minimizing the global energy $E(S; I)$: we start with an initial set of zones, either single-voxel zones or a regular grid, and greedily merging zones in order to reduce the cost.

**For each agglomeration step $t$**

Image $I$

**1.** Precompute image features

Image features
$x \mapsto \phi(x; I)$

**2.** Initialize action scores
$\Delta^e_{S^0} E(S^0 + e; I) \quad \forall e \in A^0$

Initial supervoxels $S^0$

$t \leftarrow 0$

**4.** Update action scores $\quad \forall e \in A^{t+1}$
$\Delta^e_{S^{t+1}} E(S^{t+1} + e; I)$
$\leftarrow \Delta^e_{S^t} E(S^t + e; I)$
$+ \Delta^{e,e^t}_{S^t} E(S^t + e; I)$

if $E(S^t + e^t; I) \leq E(S^t; I) + \tau$

$t \leftarrow t + 1$

**3.** Pick action $e^t$ to
minimize $E(S^t + e^t; I)$

if $E(S^t + e^t; I) > E(S^t; I) + \tau$

**5.** Stop agglomeration

Figure 6: High-level CELIS agglomeration procedure. Arrows show the flow of data (indicated by rectangles) and control (indicated by rounded rectangles). At a high-level, agglomeration proceeds in a sequential manner. At each agglomeration step $t$, the next action $e^t$ if selected to greedily minimize the global energy $E(S^t + e^t; I)$. If the best $e^t$ decreases the energy by more than $\tau$, i.e. $\Delta^{e^t}_{S^t} E(S^t; I) < \tau$, then agglomeration continues. Otherwise, agglomeration terminates. To save computation at the cost of greater memory use, the image feature vector $\phi(x; I)$ for all positions $x$ are precomputed prior to the start of agglomeration. The parallel pipeline used to initialize and update the action scores (steps 2 and 4) is shown in detail in Fig. 8; the details of the data structures that are updated by these steps are shown in Fig. 7.

# D  Implementation

A high-performance implementation of our agglomeration procedure is critical for testing and applying it to the large datasets inherent to neuronal reconstruction. The implementation challenges are, however, considerable:

- Conceptually the local search over the space of agglomerations depends on the value of an enormous number of distinct local energy terms.

- The pruning tricks described in Appendix C greatly reduce the number of shape descriptor and local energy model computations, but at the cost of significant algorithmic complexity.

- We wish to be able to use a high-dimensional image feature representation $\phi(x; I)$. Storing the precomputed 512-dimensional image features over just as a small $256^3$ voxel volume in 32-bit floating point format requires 34 GB of memory. While in absolute terms this is not a large amount of memory, it limits the number of independent volumes that may be agglomerated in parallel on a single machine, and for reasonable cost-effectiveness it is necessary, therefore, that a single agglomeration be able to take advantage of multiple cores.

- The computational steps required are not primarily standard operations like convolutions, Fourier transforms, matrix multiplications, or other linear algebra operations for which there has already been extensive study of efficient implementation techniques and for which high-performance implementations (for single and multiple CPU cores, as well as for GPU platforms) are already available.

To address these challenges, we designed a parallel pipeline that, at agglomeration step $t$, determines which shape descriptors and local energy terms need to be computed, performs those computations, and updates $\Delta^e_{S^t} E(S^t; I)$ for candidate actions $e$, in order that the action $e$ that greedily minimizes $E(S^t + e; I)$ may be chosen.

## D.1  Data structures maintained during agglomeration

This pipeline is based around several interlinked data structures, as shown in Fig. 7:

- The initial segmentation $S^0$ serves to define the agglomeration space over which our local search will operate. While conceptually we represent segmentations as an undirected graph over voxels (as described in Appendix A), we assume for simplicity that each initial supervoxel, i.e. connected component $u \in \mathcal{K}(S^0)$, is a clique, as described in Appendix C.1. This allows us to unambiguously represent $S^0$ by labeling each voxel with an integer that uniquely identifies the supervoxel that contains it. Note that it is *not* in general the case that the connected components of $S^t$ at steps $t > 0$ are cliques, meaning that we cannot unambiguously represent $S^t$ by a component labeling. In fact we do not explicitly represent the segmentation $S^t$ at later steps; instead it is represented implicitly by the initial segmentation $S^0$ and the sequence of actions $a^1, \ldots, a^t$ that have been performed.

- The global set of actions $A^t$. As described in Appendix C.1, each action $e \in A^t$ corresponds to a pair $\{u, v\} \subset \mathcal{K}(S^0)$, i.e. $e = e_{u,v}$. We represent each action $e_{u,v}$ by the pair of integer identifiers corresponding to the supervoxels $u$ and $v$. The action set at any step $t > 0$ is simply $A^0 - \{e^{t'} \mid t' < t\}$. For each action $e \in A^t$, we also maintain the set of connectivity regions $C$ in which it is active, i.e. $e \in A^t[C]$. This allows Theorem 2 to be applied efficiently. Recall that by Lemma 6, actions are removed from $A^t[C]$ during the course of agglomeration, but are never added. Thus, once an action $e$ is no longer active in any connectivity region, it ceases to affect the global energy.

- The key information that the pipeline serves to maintain is the change in global energy, $\Delta^e_{S^t} E(S^t; I)$, that would result from merging each action $e$. This change in energy is essentially a *score* associated with the action. Our agglomeration procedure follows the greedy policy of choosing at each step the action with the lowest (i.e. most negative) score. Therefore, for each active set $e$ we store the associated

Figure 7: Data structures for implementing CELIS agglomeration. Arrows indicate the links that make up the data structures.

For each connectivity region $C$, we maintain a disjoint sets data structure that maps supervoxels $u \in \mathcal{K}(S^0)$ to connected components of $K \in \mathcal{K}(S^t[C])$. For each connected component $K$, we maintain a list of incident supervoxel merge actions $e \in A^t$ to allow for efficient application of Theorem 3. This represents the multigraph obtained by contracting the connected components of $S^t[C]$. For each shape descriptor specification $s$ for the connectivity region is used, we also maintain the zone information and zone visibility sets (represented as bit vectors) for each connected component $K$.

For each action $e \in A^t$, we store $\Delta_{S^t}^e E(S^t; I)$, which serves as the ordering key for a priority queue over actions used for greedy agglomeration. We also maintain for each action $e$ the set of connectivity regions for which $e \in A^t[C]$, for efficient application of Theorem 2.

score, and we also maintain a priority queue over the scores, to allow for efficiently finding the edge with the lowest score.

- Another major component is a data structure representing the set of connectivity regions, i.e. the union of the connectivity region tilings $\mathcal{C}_s$ for each shape descriptor specification $s$.[1] The set of connectivity regions remains fixed throughout agglomeration. For each connectivity region, we maintain the following information:

  - A mapping from global supervoxels $u \in \mathcal{K}(S^0)$ (represented by unique integer identifiers) to connected components $K \in \mathcal{K}(S^t[C])$ within the connectivity region (also represented by unique integer identifiers within each connectivity region, separate from the global supervoxel identifiers). We handle the mapping of global supervoxel identifiers using a hash table, and we maintain the connected components using a standard disjoint sets data structure based on union by rank and path compression. [2, p. 505]
  - The active set $A^t[C]$ of actions that affect connectivity within $C$.
  - For each component $K \in \mathcal{K}(S^t[C])$, the set of incident actions $e \in A^t[C]$. Each incident action corresponds to a supervoxel merge of $K$ and some other component $K' \in \mathcal{K}(S^t[C])$ in $S^t[C]$. There may, however, be two distinct actions $e, e' \in A^t[C]$ that are both supervoxel merges of the same two components $K$ and $K'$. These sets of incident actions therefore correspond to the adjacency lists of the multigraph $S^t[C]/\mathcal{K}(S^t[C])$.
  - For each shape descriptor specification $s$ for which $C \in \mathcal{C}_s$ (typically there may only be one such $s$), we additionally maintain:
    * The partition $\mathcal{Z}_{s,C}$ of $X_C^s$. We represent each zone compactly as the union of disjoint rectangular regions.
    * For each component $K \in \mathcal{K}(S^t[C])$, the zone visibility set $W_s^{-1}(K; C)$ represented as a bit vector.

- Because the image feature representation $\phi(x; I)$ is typically expensive to compute, and the same feature is used for computing $E_s(x; S; I)$ for many different candidate segmentations $S$, we precompute the image features for all positions $x$ and store the feature vectors in a giant 4-D array. In practice the maximum volume size that can be agglomerated is limited by the available memory for storing the precomputed image feature array.

## D.2 Parallel pipeline for updating action scores

The pipeline for updating action scores is shown in Fig. 8. The same overall flow of control and data is used both (a) to compute the initial $\Delta_{S^0}^e E(S^0; I)$ scores for all actions $e \in A^0$ prior to agglomeration, and (b) to incrementally update the $\Delta_{S^t}^e E(S^t; I)$ scores from the prior agglomeration step by adding $\Delta_{S^t}^{e,e^t} E(S^t; I)$ to reflect the agglomeration action $e^t$ chosen. At a high level, it consists of the following operations:

- **Steps 2–6:** Preprocessing to determine the set of $(x, e)$ position/action pairs for which we must compute shape descriptors $r_s(x; S^t)$, $r_s(x; S^t + e)$, and in the incremental case $r_s(x; S^t + e^t)$ and $r_s(x; S^t + e^t + e)$. This preprocessing is where connectivity region-based pruning (Theorem 2), graph-based pruning (Theorem 3), visibility-based pruning (Theorem 4), and zone-based pruning (Theorem 6) applies.

Figure 8: Pipeline for updating CELIS action $\Delta^e_{S^t} E(S^t; I)$ scores. Arrows show the flow of data (indicated by rectangles) and control (indicated by rounded rectangles). The same pipeline is used both to compute the $\Delta^e_{S^0} E(S^0; I)$ scores *non-incrementally* (starting at 1a) at the start of agglomeration, and to *incrementally* (starting at 1b) update the $\Delta^e_{S^t} E(S^t; I)$ scores from the previous step by adding $\Delta^{e,e^t}_{S^t} E(S^t; I)$. Dashed lines indicate steps and dependencies that apply only to the incremental case. Green or red lines indicate steps and dependencies that apply only to pairwise or center-based shape descriptor specifications $s$, respectively. To limit the complexity of the diagram, the dependencies on the persistent data structures shown in Fig. 7 are omitted. In the non-incremental case (1a), the set of connectivity regions to update will be the full set $\cup_s \mathcal{C}_s$ and the set of merges to update (determined by step 2) will be the full active set $A^0[C]$. In the incremental case (1b), the zone visibility sets are updated in step 4 per Theorem 5 to reflect the merge of $K'_1$ and $K'_2$, *prior to* computing shape descriptors, to allow the conditions of Theorem 6 to be checked conveniently; the connected components (represented as disjoint sets of initial supervoxels $\mathcal{K}(S^0)$), which affect the component label map $X^s_C \to \mathcal{K}(S^t[C])$, are updated in step 13 only *after* computing shape descriptors, because the incremental update depends on computing shape descriptors $r_s(x; S^t)$ and $r_s(x; S^{t+1})$ based on both the existing and next segmentation state.

- **Step 7:** Computation of shape descriptors $r_s(x; S^t)$, $r_s(x; S^t + e)$, and in the incremental case $r_s(x; S^t + e^t)$ and $r_s(x; S^t + e^t + e)$ for the necessary $(x, e)$ position/action pairs. According to descriptor-based pruning (Theorem 1), we determine which local energy terms must be computed.

- **Steps 9–10:** Computation of local energy terms needed to compute non-zero $\Delta^e_{S^t} E_s(x; S^t; I)$ terms or, in the incremental case, non-zero $\Delta^{e,e^t}_{S^t} E_s(x; S^t; I)$ terms.

- **Steps 11–12:** Updating the global action scores based on the local energy changes.

---

**Algorithm 2** Computation of a single shape descriptor.

---

**Require:** $s$ is a shape descriptor specification.
**Require:** $\mathcal{K}$ is a set of components, represented by integers.
**Require:** $F \colon Z^3 \to \mathcal{K}$ maps shape descriptor offsets to components.
1: **function** COMPUTEDESCRIPTOR(s, F)
2:     **Declare** $|s|$-bit vector $r$
3:     **if** $s$ is pairwise **then**
4:         **for** $\{a, b\} \in s$ **do**
5:             $r^{\{a,b\}} \leftarrow \mathbb{1}_{F(a)=F(b)}$
6:         **end for**
7:     **else**
8:         $K \leftarrow F(\vec{0})$
9:         **for** $\{a, \vec{0}\} \in s$ **do**
10:           $r^{\{a,\vec{0}\}} \leftarrow \mathbb{1}_{F(a)=K}$
11:         **end for**
12:     **end if**
13:     **return** $r$
14: **end function**

---

The pipeline executes using all available processors on a single machine, through the use of a thread pool. The low-level details of the pipeline steps are as follows:

1. **(a) Before agglomeration/(b) Pick action** $e^t$ **to minimize** $E(S^t + e^t; I)$.

2. **Determine connectivity regions to update.** In the non-incremental case, all connectivity regions $C \in \cup_s \mathcal{C}_s$ must be processed. In the incremental case, per Theorem 2, only connectivity regions in $C \in \{C \in \cup_s \mathcal{C}_s \mid e^t \in A^t[C]\}$ must be processed. Because we maintain this set of connectivity regions for each action $e \in A^t$, there is only constant (low) overhead for each connectivity region *processed*, and no cost for connectivity regions not processed.

3. **Per-connectivity region processing:** The connectivity regions that must be updated are processed in parallel. While most processing is actually done at the finer per-zone granularity, certain information is computed per-connectivity-region and per associated shape descriptor $s : C \in \mathcal{C}_s$:

   - **Component label map:** a 3-D array that maps positions in the space $X_C^s$ to components in $\mathcal{K}(S^t[C])$, represented by integer identifiers. This is computed by mapping the supervoxel identifier for each position $x \in X_C^s$, which is precisely what is stored to represent $S^0$, to the corresponding component based on the map from global supervoxels $\mathcal{K}(S^0)$ to connected components $\mathcal{K}(S^t[C])$ in $C$ that we maintain.

   - $K_1', K_2' \in \mathcal{K}(S^t[C])$ **merged by** $e^t$ **(incremental only):** We also use the global supervoxel to local connected component map to translate the action $e^t$ to the pair of components $K_1', K_2' \in \mathcal{K}(S^t[C])$ for which it is a supervoxel merge. Note that it is guaranteed that $e^t$ is a supervoxel merge in $C$ because in the incremental case we only process connectivity regions $C$ for which $e^t \in A^t[C]$.

**Algorithm 3** Computation of shape descriptor changes (non-incremental case). The result is (a) a stream of position/shape descriptors pairs produced by calls to **EmitDescriptor**$(x, r)$, which returns the stream position; (b) a separate stream of score adjustments produced by calls to **EmitScoreAdjustment**$(m, i^-, i^+)$ that associate a merge $m = \{K_1, K_2\}$ with a negative and positive energy contribution corresponding to previously emitted shape descriptors at stream positions $i^-$ and $i^+$, respectively. The computation of individual shape descriptors is shown in Algorithm 2.

---

**Require:** $X \subset \mathbb{Z}^3$ is a set of positions.
**Require:** $L \colon X \to \mathcal{K}$ maps positions in $X$ to components in $\mathcal{K}$.
**Require:** $\mathcal{M} \colon \mathcal{K} \to 2^{[\mathcal{K}]^2}$ maps components in $\mathcal{K}$ to sets of merges.
1: **function** CompuTeDescriptorChanges$(s, X, L, \mathcal{M})$
2:     **Declare** array $\psi \colon \mathcal{K} \to \mathcal{K}$
3:     **for** $K \in \mathcal{K}$ **do**
4:         $\psi(K) \leftarrow K$                                          ▷ Initialize $\psi$ to the identity map.
5:     **end for**
6:     **for** $x \in X$ **do**
7:         $r \leftarrow$ CompuTeDescriptor$(s, c \mapsto L(x + c))$
8:         $i \leftarrow -1$                                    ▷ $-1$ represents an invalid index
9:         **for** $\{K_1, K_2\} \in \mathcal{M}(L(x))$ **do**
10:             $\psi(K_2) \leftarrow K_1$
11:             $r_e \leftarrow$ CompuTeDescriptor$(s, c \mapsto \psi(L(x + c)))$
12:             $\psi(K_2) \leftarrow K_2$                     ▷ Restore $\psi$ to identity map.
13:             **if** $r \neq r_e$ **then**
14:                 **if** $i = -1$ **then** $i \leftarrow$ **EmitDescriptor**$(x, r)$
15:                 $i_e \leftarrow$ **EmitDescriptor**$(x, r_e)$
16:                 **EmitScoreAdjustment**$(\{K_1, K_2\}, i, i_e)$
17:             **end if**
18:         **end for**
19:     **end for**
20: **end function**

**Algorithm 4** Computation of shape descriptor changes (incremental case).

---

**Require:** $\{K_1', K_2'\} \subseteq \mathcal{K}$ is a merge.

 1: **function** COMPUTEDESCRIPTORCHANGESINCREMENTAL$(s, \{K_1', K_2'\}, X, L, \mathcal{M})$

 2:     **Declare** arrays $\psi, \psi' \colon \mathcal{K} \to \mathcal{K}$

 3:     **for** $K \in \mathcal{K}$ **do**

 4:         $\psi(K), \psi'(K) \leftarrow K$                                 $\triangleright$ Initialize $\psi$ and $\psi'$ to the identity map.

 5:     **end for**

 6:     $\psi'(K_2') \leftarrow K_1'$

 7:     **for** $x \in X$ **do**

 8:         $r \leftarrow$ COMPUTEDESCRIPTOR$(s, c \mapsto L(x + c))$

 9:         $r_{e'} \leftarrow$ COMPUTEDESCRIPTOR$(s, c \mapsto \phi'(L(x + c)))$

10:         $i, i_{e'} \leftarrow -1$                                    $\triangleright$ $-1$ represents an invalid index

11:         **for** $\{K_1, K_2\} \in \mathcal{M}(L(x))$ **do**

12:             $\psi(K_2) \leftarrow K_1$

13:             $J_1' \leftarrow \psi'(K_1'), \; J_2' \leftarrow \psi'(K_2')$

14:             **if** $K_2 \in \{K_1', K_2'\}$ **then**

15:                 $\psi'(K_1) \leftarrow \psi'(K_2)$

16:             **else**

17:                 $\psi'(K_2) \leftarrow \psi'(K_1)$

18:             **end if**

19:             $r_e \leftarrow$ COMPUTEDESCRIPTOR$(s, c \mapsto \psi(L(x + c)))$

20:             $r_{e,e'} \leftarrow$ COMPUTEDESCRIPTOR$(s, c \mapsto \psi'(L(x + c)))$

21:             $\psi(K_2) \leftarrow K_2$                           $\triangleright$ Restore $\psi$ to identity map.

22:             $\psi'(K_1) \leftarrow J_1', \; \psi'(K_2) \leftarrow J_2'$         $\triangleright$ Restore $\psi'$ to initial value.

23:             **if** $r_e \neq r_{e,e'}$ **then**

24:                 **if** $r_{e'} \neq r_{e,e'}$ **then**

25:                     **if** $i_{e'} = -1$ **then** $i_{e'} \leftarrow$ **EmitDescriptor**$(x, r_{e'})$

26:                     $i_{e,e'} \leftarrow$ **EmitDescriptor**$(x, r_{e,e'})$

27:                     **EmitScoreAdjustment**$(\{K_1, K_2\}, i_{e'}, i_{e,e'})$

28:                 **end if**

29:                 **if** $r \neq r_e$ **then**

30:                     **if** $i = -1$ **then** $i \leftarrow$ **EmitDescriptor**$(x, r)$

31:                     $i_e \leftarrow$ **EmitDescriptor**$(x, r_e)$

32:                     **EmitScoreAdjustment**$(\{K_1, K_2\}, i_e, i)$         $\triangleright$ Note the order of $i_e$ and $i$.

33:                 **end if**

34:             **else if** $r \neq r_{e'}$ **then**

35:                 **if** $i = -1$ **then** $i \leftarrow$ **EmitDescriptor**$(x, r)$

36:                 **if** $i_{e'} = -1$ **then** $i_{e'} \leftarrow$ **EmitDescriptor**$(x, r_{e'})$

37:                 **EmitScoreAdjustment**$(\{K_1, K_2\}, i_{e'}, i)$         $\triangleright$ Note the order of $i_{e'}$ and $i$.

38:             **end if**

39:         **end for**

40:     **end for**

41: **end function**

---

- **Visibility summed area table (incremental only):** We compute a single summed area table for $K_1' \cup K_2'$ based on the component label map according to Algorithm 1.

4. **Determine the set of actions to update.** In this step, for a given connectivity region, we determine the set of actions $e \in A^t[C]$ for which me may *potentially* need to compute shape descriptors $r_s(x; S^t)$, $r_s(x; S^t + e)$, and in the incremental case $r_s(x; S^t + e^t + e)$, according to Theorem 2. Note that these actions will additionally be filtered in step 6 on a per-zone basis. In the non-incremental case, and also in the incremental case for pairwise $s$, all actions $e \in A^t[C] - \{e^t\}$ must be (potentially) processed. In the incremental case for center-based $s$, only actions $e \neq e^t$ incident to $e^t$ in $S^t[C]$ must be processed, per Theorem 3. Because we maintain the set of actions incident to each component in $\mathcal{K}(S^t[C])$, computing this set requires only constant time per action to be processed.

   **Outputs:**

   - **Merges to update:** the set $M_C$ of *merges*, i.e. pairs of components $\{K_1, K_2\} \subset \mathcal{K}(S^t[C])$ (represented as pairs of integer component identifiers) merged by the actions to be processed. Note that the same pair of components may correspond to more than one action $e \in A^t$, but computation of shape descriptors depends only on the pair of components merged by the action. We therefore use the component representation to avoid redundant computations.
   - **Merges to action map:** a mapping from each component pair in $M_C$ to the set of one or more corresponding actions:

     $$\{K_1, K_2\} \in M_C \mapsto$$
     $$\{e_{u_1, u_2} \in A^t \mid u_1 \in K_1 \wedge u_2 \in K_2\}.$$

     This is implemented as a hash table mapping pairs of component identifiers to lists of actions. Because energy terms will be locally computed per component pair rather than per action, but globally we maintain per-action scores, this mapping is used to efficiently update all corresponding global per-action scores according to each local per-component-merge score.
   - **Zone visibility sets (incremental only):** The zone visibility sets, which are represented as a mapping from integer component identifiers to bit vectors, are updated in this step per Theorem 5 to reflect the merge of $K_1'$ and $K_2'$ in $(S^t + e^t)[C]$. This simply involves taking the bit-wise OR of the bit vectors.

5. **Per-zone processing**: It is at the granularity of zones that shape descriptor computation actually happens. All zones are processed independently, and in parallel (zones of separate connectivity regions are also processed in parallel) to the extent that there are available cores. Zone processing does, however, depend on certain read-only data structures that are computed per-connectivity region and shared by all zones, including the component label map $L_s^C$, the set of merges $M_s^C$ to potentially update, and in the incremental case, the visibility summed area table.

6. **Determine set of merge/position pairs to update in zone $Z$.** The purpose of this step is to finish preprocessing in order to finalize the set of $(x, e)$ position/merge pairs for which we will compute shape descriptor changes. Per Theorem 6 and Definition 21, we filter the set of per-connectivity-region merges to update $M_s^C$ based on the zone visibility sets $M_s^Z := \{\{K_1, K_2\} \in M_s^C \mid Z \in W_s^{-1}(K_1^*; C) \cap W_s^{-1}(K_2^*; C)\}$, where in the non-incremental case $K^* := K$ but in the incremental case $K^*$ is the component $K \in \mathcal{K}((S^t + e^t)[C])$ that contains $K$. Note that in the implementation this happens transparently because the zone visibility bit vectors that are maintained for each component identifier are updated in the incremental case in step 4 to reflect $S^{t+1} = S^t + e^t$. We output either this flat merge set directly or a table of merges incident to each component in $(S^t + e^t)[C]$, depending on whether $s$ is pairwise or center-based.

   **Outputs:**

- **Positions $X_Z^s$ to update:** In the non-incremental case, the set of positions to update is simply $X_Z^s := Z$. In the incremental case, we apply Algorithm 1 to the visibility summed area table precomputed in step 3 in order to determine the subset of positions $X_Z^s \subseteq Z$ that must be updated. The time complexity is linear in $|Z|$. To limit preprocessing overhead, we only use the first condition of Theorem 4 and do not test the more complicated second condition.

- **Merge set $M_s^Z$ (pairwise $s$ only):** In the case of a pairwise shape descriptor specification $s$, we can perform no further merge pruning, and must process all merges in $M_s^Z$.

- **Component to merge set map $\mathcal{M}_s^Z$ (center-based $s$ only):** In the case of a center-based shape descriptor specification $s$, the subset of merges in $M_s^Z$ that must be processed for a given position $x$ depends on $K(x; S^t[C])$ in the non-incremental case, or $K(x; (S^t + e^t)[C])$ in the incremental case. We therefore compute a table $\mathcal{M}_s^Z : \mathcal{K}(S^t[C]) \to 2^{M_s^Z}$ that maps

$$K \in \mathcal{K}(S^t[C]) \mapsto$$
$$\big\{ \{K_1, K_2\} \in M_s^Z \,\big|\, K^* \in \{K_1^*, K_2^*\} \big\},$$

where $K^*$ is defined as above. In the non-incremental case, each merge in $M_s^Z$ will occur exactly twice in the table. In the incremental case, each merge will occur exactly 3 times in the table, because every merge in $M_s^Z$ is necessarily incident in $S^t[C]$ to $(K_1', K_2')$.

7. **Compute shape descriptors:** computation of shape descriptors $r_s(x; S^t)$, $r_s(x; S^t + e)$, and in the incremental case $r_s(x; S^t + e^t)$ and $r_s(x; S^t + e^t + e)$ for all $(x, e)$ position/merge pairs determined in step 6. To abstract the difference between pairwise and center-based descriptors, in the case of pairwise $s$, we define $\mathcal{M}_s^Z : \mathcal{K}(S^t[C]) \to 2^{M_s^Z}$ as the constant function $K \mapsto M_s^Z$. In the non-incremental case, we invoke COMPUTEDESCRIPTORCHANGES$(s, X_Z^s, L_s^C, \mathcal{M}_s^Z)$ defined in Algorithm 3. In the incremental case, we invoke COMPUTEDESCRIPTORCHANGESINCREMENTAL $(s, \{K_1', K_2'\}, X_Z^s, L_s^C, \mathcal{M}_s^Z)$ defined in Algorithm 4.

**Outputs:**

- **Action score adjustments:** the list of $\langle \{K_1, K_2\}, x, r^-, r^+ \rangle$ tuples specifying updates to the global action scores, implicitly associated with a particular shape descriptor specification $s$. Each update in the list applies to the one or more global actions that are supervoxel merges of $K_1$ and $K_2$ in $S^t[C]$, and corresponds to subtraction of $\hat{E}_s\left(r^-; \phi(x; I)\right)$ and addition of $\hat{E}_s\left(r^+; \phi(x; I)\right)$. Specifically, if we let $U$ denote the aggregate set of all action score adjustments $\langle s, \{K_1, K_2\}, x, r^-, r^+ \rangle$, then we have

$$\Delta_{S^{t+1}}^{e_{u_1,u_2}} E(S^{t+1}; I) \tag{2}$$
$$= \Delta_{S^t}^{e_{u_1,u_2}} E(S^t; I) \tag{3}$$
$$+ \sum_{\langle s, \{K_1, K_2\}, x, r^-, r^+ \rangle \in U : u_1 \in K_1 \wedge u_2 \in K_2} \left[ \hat{E}_s\left(r^+; \phi(x; I)\right) - \hat{E}_s\left(r^-; \phi(x; I)\right) \right]. \tag{4}$$

We represent the connected components $K_1$ and $K_2$ by their corresponding integer identifiers. The same shape descriptors $r^-$ and/or $r^+$ may occur in multiple action score adjustments, e.g. if they are equal to $r_s(x; S^t)$ or $r_s(x; S^t + e^t)$. To avoid redundant storage in memory and redundant evaluation of the local energy model, we do not directly specify $x$, $r^-$, and $r^+$ in our representation of the action score adjustments list. Instead, we specify $r^-$ and $r^+$ as integer offsets $i^-$ and $i^+$ into the list of **shape descriptors** and **shape descriptor positions** also output by this step.

- **Shape descriptors/Shape descriptor positions:** equal length lists specifying the non-redundant shape descriptors/position pairs required by at least one action score adjustment. The lists are constructed in such a way that the $\langle r, x \rangle$ pairs are guaranteed to be unique. The entries are grouped by position $x$, meaning that if all $\langle r, x \rangle$ pairs for a given position $x$ are contiguous.

8. **Per-batch processing of shape descriptors:** Evaluation of the local energy model on single shape descriptor/image feature pairs may be significantly more expensive than batch evaluation on multiple such pairs. For example, the matrix-vector multiplication required for typical fully-connected neural network activation can be much more efficiently implemented batch-wise as a matrix-matrix multiplication. We therefore collect the shape descriptor/position pairs output from step 7 into batches up to some maximum batch size, e.g. 256. Because different local energy models are used for each shape descriptor specification $s$, batches are segregated by specification $s$. We

9. **Extract image features.** We simply copy the image feature vectors $\phi(x; I)$ for each position $x$ in the list of shape descriptor positions for the current batch from the in-memory precomputed image feature array.

    **Output:**

    - **Image feature vectors:** temporary array holding the copied image feature vectors contiguous in memory.

10. **Compute local energy.** We evaluate in local energy terms $\hat{E}_s(r; v)$ for the current batch of shape descriptors $r$ and image feature vectors $v$.

    **Output:**

    - **Local energy terms:** the list of local energy scores corresponding to the list of shape descriptors in the current batch.

11. **Update action scores.** In this step, we update the global action scores according to Eq. (2), using the local energy terms computed in step 10 that are referenced by the action score adjustments computed in step 7. To determine the set of (global) actions that correspond to each pair of local connected components specified in the action score adjustments, we we use the merge to action map computed in step 4 for the connectivity region.

12. **Update action priority queue.** After all updates to global action scores are complete, we must correct the ordering of the action priority queue. When performing the initial action score computation prior to agglomeration, we can simply construct the heap in linear time. In the incremental case, we correct the placement of just the action for which the score was updated.

13. **Update connected components (incremental only).** In the incremental case, after computing the update action scores, we update within each affected connectivity region the disjoint sets data structure over supervoxels and the multigraph over connected components to reflect the merge $e^t$. We do not perform this update until after updating the action scores because in step 7 we need to compute shape descriptors for the segmentation states $S^t$ and $S^t + e$, which would not be possible after merging $e^t$.

# E  Performance results

We also measured the effectiveness of each of the computational pruning tricks described in Appendix C. Essentially the entire computational cost of CELIS is in computing shape descriptors and evaluating the local energy models; the cost of performing the pruning and other preprocessing turns out to be negligible (less than 1%). Therefore the savings in descriptors processed correspond directly to savings in overall runtime. With pruning, computation of shape descriptors accounted for about 20% of the cost; the remainder was spent evaluating the energy model. Without it, the cost is several orders of magnitude higher. The results are shown in Fig. 9.

Figure 9: Effect of pruning on number of shape descriptors computed. The vertical axis specifies the *cumulative* number of shape descriptors computed during the course of agglomeration, using different combinations of pruning rules. The horizontal axis specifies the agglomeration step $t$, with $t = 0$ indicating the computation required to *initialize* the energy first derivative terms. *Non-incremental* corresponds a naïve implementation that does no pruning or incremental computation whatsoever. The different combinations of *CR*, *Visibility*, *Zone*, and *Graph* correspond to correspond to applying combinations of Theorem 2, Theorem 4, Theorem 6, and Theorem 3, respectively. The actual number of descriptors that changed is shown as the lower bound, since in the best case pruning would eliminate the computation of all but these descriptors. This is also the number of evaluations of the energy model performed. If the combination of pruning techniques were perfect, it would exactly match this lower bound. Results are shown for a $100^3$ portion of the training dataset.

## Footnotes

[1]In the typical case that different tile sizes $\bar{B}_s$ and strides $\text{stride}_s$ are used for each specification $s$, these tilings $\mathcal{C}_s$ will be disjoint (but certainly overlapping, as they cover the same space), meaning that each connectivity region $C$ is associated with only one specification $s$. In general, though, there may be multiple shape descriptor specifications $s$ for which $C$ is used, i.e. $C \in \mathcal{C}_s$. Sharing a connectivity region for multiple shape descriptor specifications slightly reduces memory and computational overhead, because the per-connectivity region data structures, namely the connected components $\mathcal{K}(K)$ and active action sets $A^t[C]$, only have to be stored and maintained once.