[Reviews · NeurIPS 2016]

Reviewer 1

Summary

This paper presents an energy-based framework for segmentation of 3D volumes, and specifically targets the problem of segmenting individual neuronal processes in electron microscopy images. They use a convolutional neural network for learning a local boundary detector, a fully connected neural network for modeling an energy function over candidate segmentations, and at test time use a greedy strategy for minimizing the energy function for a given 3D image. They evaluate their method on a single dataset, a large-scale microscopy dataset containing billions of labeled voxels, and their method beats another agglomeration technique, GALA. The supplementary section of the paper presents several tricks that make their approach computationally feasible.

Qualitative Assessment

Strengths: — The approach is shown to be stronger than an existing agglomeration technique, GALA. — Aa novel shape descriptor which serves as one of the inputs to their energy function. Weaknesses: — Their approach is evaluated on a single dataset, so it is unclear how generally applicable their technique is. — Most of the techniques used in the paper (energy learning, using CNNs for detecting boundaries, greedy agglomeration etc.) are not novel. — The work falls broadly under the category of “deep structured learning”, i.e. combining deep networks (e.g. CNNs) and structured losses/models (CRFs, SSVMs, etc). Despite falling clearly in this category, the manuscript seems largely ignorant of advances in this direction: A. Schwing and R. Urtasun Fully Connected Deep Structured Networks In arXiv:1503.02351, March 2015 L. C. Chen, A. Schwing, A. Yuille and R. Urtasun Learning Deep Structured Models In International Conference on Machine Learning (ICML), Lille, France, July 2015 S. Zheng, S. Jayasumana, B. Paredes, V. Vineet, Z. Su, D. Du, C. Huang and P. Torr Conditional Random Fields as Recurrent Neural Networks International Conference on Computer Vision 2015 (ICCV 2015) Overall, this manuscript is not up to the standards of NIPS, and it seems unlikely that it will be of interest to many people other than those who are specifically working in segmenting electron microscopy images.

Confidence in this Review

2-Confident (read it all; understood it all reasonably well)


Reviewer 2

Summary

A method for segmentation of images (the domain of interest in the experiments is electron microscopy of nervous system), based on combining local information (captured by a novel shape descriptor) with oversegmentation, and optimizing over the space of possible agglomerative groupings.

Qualitative Assessment

The paper is well written (the proposed method is motivated and described clearly), the problem is important, and the results appear to be solid (in particular noting the size of the data set worked on). Let me address the claimed contributions on p. 2 in order. - "novel connectivity region data structure" -- I assume this is about connectivity regions. It's novel, although very specific to the proposed approach, and somewhat ad-hoc. - "binary shape descriptor" -- indeed novel, and to me the most interesting component here. - "a neural network architecture [...]" -- I am not sure what's novel here; e.g., [c] Semantic Image Segmentation with Deep Convolutional Nets and Fully Connected CRFs, ICLR 2015, Chen et al. - "a training procedure [...]" -- again, not sure what's novel. Pre-training pairwise potential in a CRF using pairwise affinities is quite standard. - "an experimental evaluation[...]" -- I don't have the domain expertise but while the experimental setup seems solid, I am not sure the evaluation experiment for a proposed model qualifies as a "contribution" Most of my concerns with novelty above stem from the same issue I mention as a fatal flow earlier: an almost complete lack of reference to (and, one assumes, awareness of) a vast body of literature on related topics in computer vision.

Confidence in this Review

2-Confident (read it all; understood it all reasonably well)


Reviewer 3

Summary

The paper describes a new super-voxel agglomeration scheme for segmenting neurons from large 3d electron microscopy images of brain tissue. They contribute a nice new feature representation for segmentations, a neural network based global energy function describing the goodness of a candidate segmentation which is trained to enable local optimization, and extensive evaluation on a very large ground truth volume.

Qualitative Assessment

The binary segmentation-based feature descriptors seem novel and useful. The neural net based global energy function seems quite powerful, but expensive to evaluate and optimize. By analogy to RL, this seems more like a value function, than an action-value Q-function, which might potentially be more efficient to use. The evaluations seem thorough, and are definitely on one of the largest volumes of ground truth. It would be nice to include the quantitative results of the experiments on "18 non-overlapping 500^3 subvolumes" described in Section 8 in figure 3. It's a nice paper introducing a novel method for agglomerating super-voxels regions. The main question that I'd like to see answered is: how would a much larger convnet perform? The convnet used in this paper is rather small, with only a field of view of 35x35x9. Much larger convnets are now easy to train, how well would a 3d-CNN+watershed perform, if the convnet is as large as the effective field of view of 3d-CNN+CELIS of 35x35x9 + 33x33x33?

Confidence in this Review

3-Expert (read the paper in detail, know the area, quite certain of my opinion)


Reviewer 4

Summary

Volumetric segmentation of neuronal tissue is addressed in this paper. The approach can be broadly described as energy minimization with learned CNN features. The energy is employed to determine merging of adjacent voxels. To tackle the highly complex combinatorial problem one faces, the paper also introduces binary descriptors to efficiently represent the 3D configuration. As an additional contribution, a new very large volumetric dataset is collected and annotated. Experimental results on this datasets shows reasonable performance improvement with respect to state of the art methods.

Qualitative Assessment

First, on the positive aspects. The results seem to outperform existing methods on the introduced dataset and the dataset itself is a relevant contribution. On the other hand, it could have also been possible, or maybe even desirable, to compare the performance on existing datasets. The authors argue that the newly introduced one is so extremely large that it is representative of any other experiment. However, I do not see a reason not to include one anyway. One thing I missed is a discussion on the optimality of the achieved solution. Is it at all possible to get a quantitative assessment on the achieved bounds at all or is one left in the dark concerning the optimization quality? The proposed binary shape descriptor reminded me somewhat of BRIEF (Colonder et al., ECCV 2010). Is it possible to add a note in the related work on that work? Finally, I found the technical section very hard to follow. Unfortunately, I do not have any suggestion on how to improve it, but the amount of math notation inline in the text makes the read flow rather tedious.

Confidence in this Review

1-Less confident (might not have understood significant parts)


Reviewer 5

Summary

The paper makes use of NN to model the Combinatorial Energy Learning for Image Segmentation.

Qualitative Assessment

1. Section 4 is mainly for the energy model learning Ê. According to the main equation in subsection 4.1, Ê is also defined as predicted probability. However, there is no equation to show how to compute Ê 2. Similar for other functiosn and parameters (\phi, r_s, loss function l, etc) in the paper, they are no well defined. For example, the binary shape descriptor r is mentioned in section 2, and defined in section 3. however, the one defined in section 3 is r^i(U) where as the one in the Section 2 is r_s(x; S). After the Section 3, r_s(x, S) is not defined how to compute 3. The authors should compare with some state of the art instead the one already published 3 years ago. Since the paper makes use of NN many step, the authors should compare with proposed method against the CNN-based segmentation

Confidence in this Review

2-Confident (read it all; understood it all reasonably well)


Reviewer 6

Summary

This paper proposed a novel method to handle the large-scale segmentation problem in mapping neuroanatomy. The proposed method uses deep neural network to evaluate local energy given a segmentation configuration, as well as an energy minimization approach to greedily guide the (super)voxel agglomeration. To handle the inherent large scale of the targeted problem, the authors also consider the issue of pruning from several aspects.

Qualitative Assessment

The paper is well-written, with lots of details provided. The studied problem is definitely of great impact and would significantly benefit the community of neuroscience. The reviewer noticed the large size of the problem and appreciates the considerable amount of works achieved, as described by the paper. On the other hand, I also want to point out some associated weakness: 1. To my understanding, two aspects which are the keys to the segmentation performance are: (1) The local DNN evaluation of shape descriptors in terms of energy, and (2) The back-end guidance of (super)voxel agglomeration. Although experiment showed gains of the proposed method over GALA, it is yet not clear enough which part is the major contributor of such gain. The original paper of GALA used methods different from this paper (3D-CNN) to generate edge probability maps. Is the edge map extraction framework in this paper + GALA a fair enough baseline? It would be great if such edge map can also be visualized. 2. The proposed method to some extent is not that novel. The idea of generating or evaluating segmentation masks with DNN has been well studied by the vision community in many general image segmentation tasks. In addition, the idea of greedily guiding the agglomeration of (super)voxels by evaluating energies pretty much resembles many bottom-up graph-theoretic merging in early segmentation methods. The authors, however, failed to mention and compare with many related works from the general vision community. Although the problem general image segmentation somewhat differs from the task of this paper, showing certain results on general image segmentation datasets (like the GALA paper) and comparing with state-of-the-art general segmentation methods may give a better view of the performance of the proposed method. 3. Certain parts did not provide enough explanation, or are flooded by too much details and fail to give the big picture clearly enough. For example in Appendix B Definition 2, when explaining the relationship between shape descriptor and connectivity region, some graphical illustrations would have helped the readers understanding the ideas in the paper much easier and better. ----------------------------------------------------------------------------- Additional Comments after Discussion Upon carefully reading the rebuttal and further reviewing some of the related literature, I decide to down-grade scores on novelty and impact. Here are some of the reasons: 1. I understand this paper targets a problem which somewhat differs from general segmentation problems. And I do very much appreciate its potential benefit to the neuroscience community. This is indeed a plus for the paper. However, an important question is how much this paper can really improve over the existing solutions. Therefore, to demonstrate that the algorithm is able to correctly find closed contours, and really show stronger robustness against weak boundaries (This is especially important for bottom up methods), the authors do need to refer to more recent trends in the vision community. 2. I noticed one reference "Maximin affinity learning of image segmentation, NIPS 2009" cited in this paper. The paper proposed a very elegant solution to affinity learning. The core ideas proposed in this paper, such as greedy merging, Rand Index like energy function show strong connections to the cited paper. The maximin affinity is basically the weakest edge along a minimum spanning tree (MST), and we know greedy region merging is also based on cutting weakest edges on a MST. The slight difference is the author proposed the energy term at a lot of positions and scales but the previous paper has a single energy term for the global image. In addition cited paper also addressed a similar problem in the experiment. However, the authors not only did not include the citation as an experimental baseline, but also failed to provide detailed discussions on the relation between the two works. 3. The authors argued for greedy strategy, claiming this is better than factorizable energies. "By sacrificing factorization, we are able to achieve the rich combinatorial modeling provided by the proposed 3d shape descriptors." I guess what the authors mean is they are putting more emphasis on local predictions. But this statement is not solidly justified by the paper. In addition, although to some extent I could understand this argument (local prediction indeed seems important because the size of cells vs volume are much smaller than segments vs whole image in general segmentation), there are significant advances on making pretty strong local predictions from the general vision community using deep learning, which the authors fail to mention and compare. I think overall the paper addressed an interesting problem and indeed showed solid research works. However it will better if the authors could better address contemporary segmentation literature and further improve the experiments.

Confidence in this Review

3-Expert (read the paper in detail, know the area, quite certain of my opinion)